# The potential of global coastal flood risk reduction using various DRR measures

Eric Mortensen[1], Timothy Tiggeloven[1], Toon Haer[1], Bas van Bemmel[2], Dewi Le Bars[3], Sanne Muis[1,4], Dirk Eilander[1,4], Frederiek Sperna Weiland[4], Arno Bouwman[2], Willem Ligtvoet[2], Philip J. Ward[1,4]

[1]Institute for Environmental Studies, Vrije Universiteit Amsterdam, Amsterdam, 1081 HV, The Netherlands
[2]Planbureau voor de Leefomgeving, The Hague, 2500 GH, The Netherlands
[3]Koninklijnk Nederlands Meteorologisch Instituut, De Bilt, 3731 GA, The Netherlands
[4]Deltares, Delft, 2629 HV, The Netherlands

*Correspondence to* Eric Mortensen (eric.mortensen@vu.nl)

**Abstract.** Coastal flood risk is a serious global challenge facing current and future generations. Several disaster risk reduction (DRR) measures have been posited as ways to reduce the deleterious impacts of coastal flooding. On the global scale, however, efforts to model the future effects of DRR measures (beyond structural) are limited. In this paper, we use a global-scale flood risk model to estimate the risk of coastal flooding, and to assess and compare the efficacy and economic performance of various DRR measures, namely: dykes and coastal levees, dry-proofing of urban assets, zoning restrictions in flood-prone areas, and
management of foreshore vegetation. To assess the efficacy of each DRR measure, we determine the extent to which they can limit future flood risk as a percentage of regional GDP to the same proportional value as today (a *relative-risk constant* objective). To assess their economic performance, we estimate the economic benefits and costs of implementing each measure. If no DRR measures are implemented to mitigate future coastal flood risk, we estimate expected annual damages to exceed $1.3 trillion by 2080, directly affecting an estimated 11.5 million people on an annual basis. Low- and high-end scenarios
reveal large ranges of impact uncertainty, especially in lower-income regions. At the global scale, we find the efficacy of dykes and coastal levees in achieving the *relative-risk constant* objective to be 98%; of dry-proofing to be 49%; of zoning restrictions to be 11%; and of foreshore vegetation to be 6%. In terms of direct costs, the overall figure is largest for dry-proofing ($151 billion) and dykes and coastal levees ($86 billion), much more than those of zoning restrictions ($27 million) and foreshore vegetation ($366 million). These two more expensive DRR measures also exhibit the largest potential range of direct costs.
While zoning restrictions and foreshore vegetation achieve the highest global benefit-cost ratios, they also provide the smallest magnitude of overall benefit. We show that there are large regional patterns in both the efficacy and economic performance of modelled DRR measures that display much potential for flood risk reduction, especially in regions of the world that are projected to experience large amounts of population growth. Over 90% of sub-national regions in the world can achieve their *relative-risk constant* targets if at least one of the investigated DRR measures is employed. While future research could assess
the indirect costs and benefits of these four and other DRR measures as well as their subsequent hybridisation, here we demonstrate for global and regional decision makers the case for investing in DRR now to mitigate future coastal flood risk.

## 1 Introduction

Coastal floods have historically been, and currently are, one of the deadliest and most damaging natural hazards in the world (Idier et al., 2020; Guha-Sapir et al., 2015). Sea-level rise will cause this hazard and subsequent impacts to increase in severity and affect society more frequently in future decades (Nicholls & Cazenave, 2010; Jevrejeva et al., 2018; Vitousek et al., 2017; Muis et al., 2020). Numerous studies have been conducted using global-scale models to assess coastal flood risk, its components – i.e. hazard, exposure, and vulnerability (Kron, 2005; UNDRR, 2015a), and the magnitude of impacts that are possible if this changing risk is left unaddressed (e.g., Hoozemans et al., 1993; Hinkel & Klein, 2009; Hinkel et al., 2010; Jongman et al., 2012; Hallegatte et al., 2013; Hinkel et al., 2014; Neumann et al., 2015; Schuerch et al., 2018; Vafeidis et al., 2019; Tiggeloven et al., 2020; Menéndez et al., 2020). The aforementioned studies project increases in global figures for people affected by coastal flooding, and the associated expected monetary damages, through the end of the 21$^{st}$ century. Projected increases to Expected Annual Damages (EAD) range from tens-of-billions to trillions, while increases in Expected Annual Affected Population (EAAP) are projected into the tens-of-millions.

These projected changes in coastal flood risk illustrate that the need for disaster risk reduction (DRR) measures is increasingly urgent (Griggs & Reguero, 2021; UNDRR, 2020). Forward-looking DRR (i.e. prospective disaster risk management) examines potential future risks under climate change scenarios, as does climate change adaptation. There is a call in policy-geared and scientific literature to bridge the silos between these domains (UNDRR, 2020), as well as numerous global calls for meaningful action towards reducing coastal flood risk in general, including the Warsaw International Mechanism for Loss and Damage Associated with Climate Change Impacts (UNFCCC, 2014) and the Sendai Framework for Disaster Risk Reduction (UNDRR, 2015b). The United Nations' most recent Adaptation Gap Report concluded that more ambition is needed to avert future disasters (Neufeldt et al., 2021). Disaster risk reduction is a key component to filling this gap.

Several DRR measures, such as structural measures (dykes, coastal levees, etc.), individual measures (floodproofing, flood insurance, etc.), community-wide measures (land use policy, early warning systems, etc.), and nature-based solutions (mangroves, wetlands, beach and dune nourishment, etc.), have been posited as methods to reduce the deleterious impacts of coastal flooding (Aerts et al., 2014). Some of these measures have subsequently been assessed using global models. For example, Hinkel and Klein (2009), Lincke and Hinkel (2018), Tamura et al. (2019), Tiggeloven et al. (2020), and Schinko et al. (2020), all improve the understanding of structural DRR measures now and in the future, demonstrating that large benefits can be achieved by raising current flood protection levels.

Other DRR measures, such as building-level protection, nature-based solutions, or zoning restrictions, meanwhile, have received much less attention in global-scale literature. Regarding building-level protection, several studies have assessed floodproofing of buildings against river flooding up to the national or continental level (e.g., Kreibich et al. (2005) and Haer et al. (2019)), but such studies are so far missing for coastal flooding at the global scale. In terms of nature-based solutions,

some studies have quantified the current flood risk reduction benefits of conserving existing natural areas (Tiggeloven et al., 2022; Menéndez et al., 2020; Beck et al., 2018), but only Beck et al. (2022) do so regarding the future benefit of expanding these ecosystems, and only for the Caribbean. Hinkel and Klein (2009) assessed the effect of beach nourishment and wetland nourishment on coastal flood risk. To date, no studies have assessed the global-scale potential of zoning restrictions, although local case studies such as Muis et al. (2015) for Java and Koks et al. (2014) for Belgium point towards the potential of expanding the modelling of this DRR measure to the global scale. Furthermore, there is a distinct lack of studies that have compared the efficacy, which we define here as the performance of any given DRR measure under ideal and controlled circumstances, and/or economic performance of different DRR measures on the global scale.

The overarching aim of this paper is therefore to assess and compare the efficacy and economic performance (in terms of economic benefits and costs) of various DRR measures on coastal flooding at the global scale. We assess four different DRR measures, namely: 1) dykes and coastal levees; 2) foreshore vegetation; 3) zoning restrictions; and 4) dry-proofing of buildings. To assess the efficacy of each DRR measure, a baseline of desired risk reduction must be established. Different criteria have been considered to establish what level of risk reduction should be aimed for and subsequently compared (Tiggeloven et al., 2020). As outlined by the Sendai Framework, an equitable target of risk reduction needs to be established to ensure a realistic pathway forward for all regions of the world, regardless of development state or socioeconomic status. Here, we use a target in which future flood risk (explicitly, EAD) as a percentage of regional GDP remains constant in the future at the same level as today, also known as the *relative-risk constant* objective, as described in Ward et al. (2017) and Tiggeloven et al. (2020).

Two main research questions guide this work: 1) what level of risk reduction is needed to achieve the targets set by a *relative-risk constant* objective; and 2) to what extent can various DRR measures achieve this level of risk reduction? In our analysis, because the measures are implemented to protect against a quantifiable return period of inundation in the future, we use the term DRR to refer to any actions taken to address changes in coastal flood risk. To answer these questions, we expand GLOFRIS, a global flood risk estimation methodology developed for coastal flooding (Ward et al., 2013; Tiggeloven et al., 2020), by implementing several DRR measures and assessing their efficacy against a pre-defined risk target. In our analysis, we have developed and modelled dry-proofing and zoning restrictions as DRR measures. This has never been done before on the global scale (Ward et al., 2015). We have also incorporated previously unconsidered costs for foreshore vegetation, namely mangrove restoration costs. To fully compare these new findings, the impact reduction potential of dykes and coastal levees as well as foreshore vegetation are recalculated. We have conducted this study using new hazard and exposure data that were developed explicitly for this analysis.

## 2 Methodology

The methodological components of this paper can be summarised in three steps (Figure 1): (1) risk estimation (section 2.1); (2) simulating risk reduction (section 2.2); and (3) measure evaluation (section 2.3). In the first step, we estimate current and future coastal flood risk by combining data on hazard, exposure, and vulnerability, and taking the integral of the resulting risk curve to calculate EAD and EAAP. We also set a risk reduction target per sub-national region (GADM, 2020) using the *relative-risk constant* objective. In the second step, we estimate the risk reduction that could be achieved by each of the DRR measures studied as well as the residual risk remaining after taking these measures. The colours displayed in Figure 1 relate to the individual components of coastal flood risk that we model and assess: blue for hazard (reduction), green for exposure (reduction), and purple for vulnerability (reduction). In the third step, we determine the efficacy of each measure in terms of the extent to which they can achieve the *relative-risk constant* risk reduction target and perform a benefit-cost analysis.

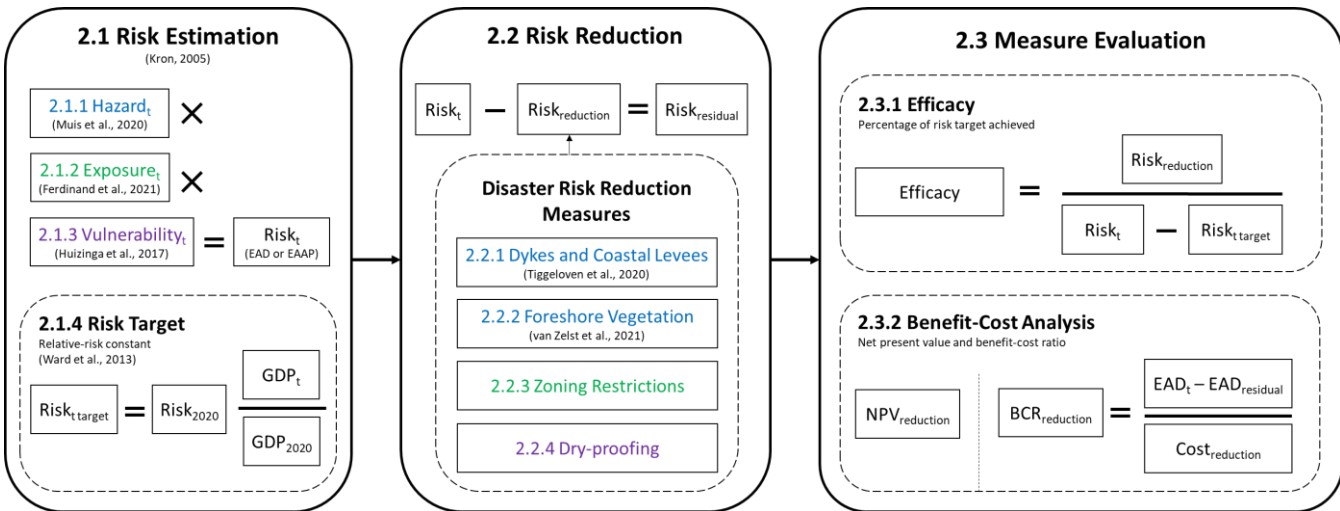

**Figure 1: Our study's analysis follows three main methodological steps consisting of estimating current and future coastal flood risk (labelled as "2.1 Risk Estimation"), modelling of four distinct DRR measures ("2.2 Risk Reduction"), and evaluating said measures in terms of their efficacy and economic performance ("2.3 Measure Evaluation"). Subsequent numbering in the figure refers to the relevant subsection of text. The literature referenced in the figure denotes the source of data or methodology used (as described in further detail in subsequent sections), while the colours represent different components of risk addressed.**

### 2.1 Risk estimation

We extend the GLOFRIS framework to estimate risk in our study. GLOFRIS was originally developed for global riverine flood risk modelling (Ward et al., 2013; Winsemius et al., 2013) and recently modified for global coastal flood risk modelling (Tiggeloven et al., 2020). We expand the framework by adding several DRR measures and a module for assessing their efficacy against a pre-defined risk reduction target. GLOFRIS simulates flood risk by combining information on hazard, exposure, and vulnerability. Hazard is represented by inundation maps showing the flood extent and depth for different return periods at a

horizontal resolution of 30" x 30" under various climate scenarios. Exposure is represented by population maps and land use maps that show urban areas and estimate their economic value (also 30" x 30"). Vulnerability is represented by so-called vulnerability curves, or depth-damage functions, which show the percentage of the economic value per grid cell that would be damaged at different flood depths. The datasets used to represent global coastal flood hazard, exposure, and vulnerability in this paper are discussed in sections 2.1.1, 2.1.2, and 2.1.3, respectively.

After calculating the impacts for different return periods, risk is calculated by taking the integral of the exceedance probability–impact curve (Meyer et al., 2009). EAAP and EAD values can be calculated for current conditions as well as future conditions for each DRR measure under various Representative Concentration Pathways (RCPs) and Shared Socioeconomic Pathways (SSPs). Current levels refer to those in the year 2020, while a future timestep consists of the window of 20 years on either side of a given year. We thoroughly examine future results under the 50th percentile of RCP 6.0 / SSP2 in the year 2080 in this analysis. RCP 6.0 / SSP2 is chosen here as a climate change adaptation scenario due to its representation of an intermediate level of climate change mitigation, moderate socioeconomic challenges, and a balanced emphasis on economic development, social equity, and environmental sustainability, thus presenting a realistic scenario for decision and policymakers to consider. To represent low- and high-end scenarios, we also refer to additional results (available as a dataset in our manuscript's repository) of the combinations RCP 2.6 / SSP1 and RCP 8.5 / SSP3 to further contextualise our main findings.

### 2.1.1 Hazard

This analysis uses hazard maps that show coastal floods for the following return periods: 2, 5, 10, 25, 50, 100, 250, 500, and 1000 years. The hazard maps show the extent and depth of flooding per grid cell (30" x 30") for each return period under current conditions and under sea-level rise projections (Garner et al., 2022). The coastal hazard maps were developed using data from CODEC (Muis et al., 2020), which is based on the hydrodynamic Global Tide and Surge Model (GTSM) forced with wind and pressure fields derived from ERA5 (Hersbach et al., 2020) and enriched with simulated tropical cyclones using the IBTrACS (International Best Track Archive for Climate Stewardship) archive.

We follow the Peak Over Threshold (POT) method and we fit the Generalised Pareto Distribution (GPD) on the peaks that exceed the 99th percentile surge level. From there we derive estimated sea levels for various return periods. These computed sea levels are then used as input for a GIS-based inundation model using the MERIT (EGM 96) DEM (Yamazaki et al., 2018) with spatially varying offset between mean sea level according to the FES2012 model, as described in Tiggeloven et al. (2020), in order to simulate global inundation using the same datum. This is a static flood model that simplifies all dynamic processes into a single attenuation factor of the water levels over land (Vafeidis et al., 2019), resulting in a simple bathtub model with static forcings instead of a more complex dynamic inundation model framework. The flood maps do not include future changes in waves and storminess. Rather, (nearshore) waves are accounted for in calculation of the hazard-specific DRR measures effect on inundation levels, discussed further in sections 2.2.1 and 2.2.2 below.

Similarly to Tiggeloven et al. (2020), we include subsidence in the estimation of future coastal flood risk. These subsidence rates are determined by the SUB-CR model (Kooi et al., 2018). Because subsidence is a highly regional phenomenon, rates of subsidence are applied to specified locations of Kooi et al. (2018) at a spatial resolution of 30" x 30" (which is a spatial interpolation based on the original spatial scale of 5' x 5').

Results of RCP 6.0 (50[th] percentile), an intermediate climate change scenario, are explored here in the main text. Additional
RCPs, namely RCP 2.6 (50[th] percentile) and RCP 8.5 (50[th] percentile), are used in combination with relevant SSPs (SSP 1 and SSP3, respectively) to represent low- and high-end scenarios for the results we present here. These values are available as a deposited dataset in our study's data repository, where one may also find the hazard input described in this subsection.

### 2.1.2 Exposure

Exposure input includes current and future population and urban areas, current and future GDP, and maximum economic
damage per urban grid cell. Gridded data for population, urban areas, and GDP are taken from the 2UP model (Andree & Koomen, 2017). The 2UP model uses the Global Human Settlement (GHSL) framework (Corbane et al., 2019) as a starting point for current exposure and then independently simulates future expansion and development. GHSL provides a built-up area grid derived from Sentinel-2 global image composite for reference year 2018 using Convolutional Neural Networks (GHS-S2Net) at a resolution of 10'x10', which is resampled to 30"x30" (Ferdinand et al., 2021). Here, built-up area refers to all
kinds of built environs (e.g., buildings) and artificial surfaces (e.g., paved surfaces). Future simulations of built-up area include five narrative descriptions of future societal development associated with SSPs, which vary from sustainable global development to regional conflict (O'Neill et al., 2017). We present our results here in the main text using SSP2, or the "Middle of the Road" scenario. SSP1 ("Sustainability") and SSP3 ("Regional Rivalry"), meanwhile, are combined with relevant RCPs (RCP 2.6 and RCP 8.5, respectively) to represent low- and high-end scenarios for the results. These values are available as a
deposited dataset in our study's data repository, where one may also find the exposure input described in this subsection.

Maximum economic damages are estimated using the methodology of Huizinga et al. (2017), where a root function is used to link GDP per capita to construction costs for each country. To convert construction costs to maximum damages, several adjustments are carried out using the suggested factors by Huizinga et al. (2017) for the different occupancy types. The urban grid cells of the layers from the 2UP database are assumed globally to be 75 % residential, 15 % commercial, and 10 %
industrial, based on a European-scale study by Economidou et al. (2011) and a comparison of European cities' share of occupancy type of CORINE Land Cover data (EEA, 2016). Following Huizinga et al. (2017), the density of buildings per occupancy type are set to 20 % for residential and 30 % for commercial or industrial.

To estimate future maximum damages, the current values are scaled with the GDP per capita per country from the SSP database. Boundaries of countries and sub-national regions are derived from version 3.6 of the Global Administrative Areas

dataset (GADM, 2020). Future gridded GDP values are taken from van Huijstee et al. (2018), which uses the national GDP per capita from the SSP database as input.

### 2.1.3 Vulnerability

Vulnerability is represented using different global flood depth–damage functions (Nasiri et al., 2016) for each occupancy type mentioned in the previous section. The curves used here are taken from Tiggeloven et al. (2020) and are based on the empirical

curves described in Huizinga et al. (2017). The resulting damages are represented as a percentage of the maximum damage, or the total assumed economic value of the given cell. This maximum damage is reached at a water level depth of 6m.

### 2.1.4 Risk target

To assess the efficacy of the DRR measures, we establish a baseline risk reduction target against which each measure is benchmarked. While this target could be set in terms of financial or human impact using many indicators, in this analysis the

target risk reduction is set by the so-called *relative-risk constant* objective as defined by Ward et al. (2017) and applied by Tiggeloven et al. (2020). Specifically, we define the *relative-risk constant* target as the level of future risk in which the percentage of future impacted GDP to future total GDP is held constant to the percentage of current GDP impacted to current total GDP. The *relative-risk constant* target is calculated for each sub-national region individually, and for this study is presented at the time-step of the year 2080.

An estimate of the current flood protection standard for each sub-national region is required to calculate the current risk. We use estimates of the current protection level by applying the FLOPROS modelling approach as originally described in Scussolini et al. (2016) and applied for coastal flooding by Tiggeloven et al. (2020). In the absence of information regarding the protection standards of certain sub-national regions, a 2-yr protection standard in 2020 is assumed. Meanwhile, a maximum of a 1000-yr protection standard is assumed in only some special cases (e.g., The Netherlands and Singapore). Due to sea-level

rise and subsidence, the protection standard of current flood protection infrastructure will decrease in the future if no improvements are made (e.g., a DRR measure that provides protection against a 100-yr event today may only protect against a 75-yr event in the future). Protection standards are assumed to degrade no further than the 2-yr protection standard.

### 2.2 Risk reduction

The DRR measures modelled in this analysis can be roughly divided into three categories; measures that either reduce hazard

(i.e., dykes and coastal levees and foreshore vegetation), exposure (i.e., zoning restrictions), or vulnerability (i.e., dry-proofing). In the case of foreshore vegetation, zoning restrictions, and dry-proofing, spatial limits exist for the amount of action that can be taken – only so much area can be restored to mangroves, restricted from urbanisation, or dry-proofed, depending on the context. Meanwhile, we have modelled dykes and coastal levees with no physical limitation, such as a height limitation

or a maximum protection standard provided (aside from the theoretical limit of the 1000-year flood event). Of course, these four DRR measures could arguably influence and alter another component of risk – additional development occurring behind a dyke thus increasing potential exposure in the event of a failure (Haer et al., 2020), for example. For the purpose of simplifying the study, though, we do not explore these unintended effects or consequences of DRR measures. In the following subsections, we describe each of the four DRR measures assessed in this study.

### 2.2.1 Dykes and coastal levees

The modelling of dykes and coastal levees largely follows the methodology detailed in Tiggeloven et al. (2020), in which inundation depth is used as a proxy for dyke height and thereafter translated into a protection standard. Similar to the flood hazard modelling methodology, water levels for dyke heights are derived from GTSM (Muis et al., 2020) and corresponding current wave conditions at different return periods from the ERA-Interim reanalysis (Dee et al., 2011).

Assuming that the incoming wave direction is perpendicular to the coast, the wave attenuation over the foreshore is determined via a lookup table consisting of numerical modelling results (Tiggeloven et al., 2022; van Zelst et al., 2021). Current dyke heights with respect to local hydrological conditions, wave attenuation, and extreme sea level are calculated with the empirical EuroTop formulations (Pullen et al., 2007) and are based on a standard 1:3 dyke profile without berms. This is representative of a low-cost dyke. To calculate future dyke heights, sea-level rise is added directly to the crest height (Jackson & Jevrejeva, 2016). Subsidence levels, taken from Kooi et al. (2018), are also considered when calculating dyke height. Subsidence is assumed to take place directly on the dyke and therefore computed on the crest height, which is similar for sea-level rise calculations.

The difference between current and future dyke and coastal levee heights, multiplied by the length required and the unit costs, represents the cost associated with this DRR measure. This cost methodology is also described in Tiggeloven et al. (2020). The costs of raising dykes are estimated by calculating the total length of dyke heightening per grid cell and multiplying by a unit cost set to $7 million km m−1 based on reported costs in New Orleans (Bos, 2008). This unit cost is multiplied by a construction index multiplier, based on civil-engineering constructions costs, to adjust the construction costs to account for differences between countries (Ward et al., 2010). The unit cost value used is within a reasonable range when compared to various studies (Aerts et al., 2013; Jongman et al., 2012; Lenk et al., 2017; Tiggeloven et al., 2020). The costs are discounted with a discount rate of 5 % until 2100 (lifespan of investment) and with operation and maintenance costs of 1 %. We assume that initial investments are made in 2020, with construction complete after 20 years (Tiggeloven et al., 2020). We also assume that the dykes as modelled do not fail for water levels below the crest level and fail completely for water levels that are higher.

### 2.2.2 Foreshore vegetation

The enhanced effect of foreshore vegetation (i.e., salt marsh conservation and mangrove restoration) on wave attenuation is modelled based on the methodology of Tiggeloven et al. (2022). Similar to Tiggeloven et al., (2022), here wave conditions are derived from the ERA-Interim (Dee et al., 2011) reanalysis using a peak-over-threshold approach. To determine the wave attenuation over a foreshore and the resulting significant wave height relevant for the flood protection on a transect, we search an existing lookup-table (van Rooijen et al., 2016) of hydrodynamic numerical modelling results for combinations of foreshore slopes, vegetation covers and hydrodynamic conditions (van Zelst et al., 2021). These searched wave heights are modelled at regular intervals along a steady slope, both with and without salt marsh or mangrove vegetation. Wave angle of incidence is assumed coast normal. Wave attenuation along the vegetated coastlines is determined based on the closest match between the derived transects characteristics and look-up table results. Specifically in this research, current extents of salt marshes are conserved (Tiggeloven et al., 2022) and mangrove extents are expanded based on the Mangrove Restoration Potential Mapping Tool as developed by Worthington & Spalding (2018).

Critically, as there is no existing database of potentially restorable salt marshes, we assume a constant extent of this specific type of foreshore vegetation from now into the future. The type of vegetation is determined by maps of global salt marshes (Mcowen et al., 2017) and mangroves (Giri et al., 2011). Vegetation data from maps of the Corine Land Cover for Europe only and GlobCover v2.2 for the rest of the world are derived to complement areas of missing data.

No direct costs are considered to be incurred with preservation of existing salt marshes (Tiggeloven et al., 2022). We therefore assume that the direct costs of the foreshore vegetation DRR measure stem from mangrove restoration. The total area of new mangroves that are capable of being restored is just under one million hectares, as identified by the Mangrove Restoration Potential Mapping Tool (Worthington & Spalding, 2018). We set initial investment costs to ~$42,800 per hectare of new mangroves in high- and upper-middle-income countries and ~$1,410 in lower-middle- and low-income countries, with maintenance costs set to 2.5% of the initial investment per year (Aerts, 2018; Bayraktarov et al., 2016; Marchand, 2008; Narayan et al., 2016). We assume that all mangrove restoration, if supported and funded properly, can occur rather quickly (Ellison et al., 2020; Kaly & Jones, 1998) and is therefore complete by 2030. Throughout the remainder of the simulation, restored mangrove areas are held spatially constant.

### 2.2.3 Zoning restrictions

With zoning restrictions, we assess the potential reduction in flood risk that could be achieved by restricting future urban development in flood-prone regions. This is achieved by performing a simulation of future urban expansion using the 2UP model in which no expansion of urban areas is allowed in cells that are inundated by the 1000-yr return period flood in 2080, according to the flood hazard maps described in section 2.1.1. Cells that are classed as urban in the current time period remain

urban in the future. In 2UP, this means that the future urban cells that would otherwise develop over time in the 1000-yr flood zone are instead reassigned to another likely non-flood-prone location within the same country, based on simple suitability functions (Ferdinand et al., 2021).

265   Although it does not eliminate existing risk, this DRR measure results in fewer people and assets being exposed to flood risk under future scenarios, and therefore results in benefits as we define them in this study. The main direct cost of exposure reduction through zoning restrictions is the administrative cost associated with implementing such a policy, which varies significantly on local scales and in styles of implementation (Dunham, 1959; Porse, 2014). When compared to the costs of other measures such as dykes and coastal levees, though, the direct costs associated with zoning restriction are marginal. Here,

270   a nominal total cost per sub-national region of $2,000,000 (for high- and upper-middle-income regions) or $500,000 (for lower-middle- and low-income regions) is applied evenly over all years of simulation (Meng, 2021; Ran & Nedovic-Budic, 2016; de Bruin et al., 2014). While having minimal direct cost in its implementation, altering development with zoning restrictions also produces an opportunity cost, which is the difference of economic value of land that would have been developed within a flood-prone area as opposed to leaving that land in a less developed state (Parsons & Wu, 1991). These opportunity costs may

275   be significant on the local scale. For our analysis, though, we assume that any potential GDP growth still occurs in-country but is displaced away from the flood zone.

### 2.2.4 Dry-proofing

For vulnerability reduction, the analysis focuses on dry-proofing for individual buildings. Dry-proofing makes a structure and its contents less likely to face damage from a flood event (i.e., waterproofed) up to a certain water level (Zevenbergen et al.

280   2007). Dry-proofing, for example, has been shown to be (cost-) effective in certain regions around the world (Kreibich et al., 2005; Haer et al., 2019). The uptake of such measures can be reinforced by building codes. Recent studies have investigated this vulnerability reduction measure on a European scale by Haer et al. (2019), but a global analysis is lacking. For this global analysis, we assume that assets cannot be dry-proofed safely above 1m as water pressure above these levels would reduce the structural integrity of a building (FEMA, 2021).

285   We also assume that all assets – residential, commercial, and industrial – receive this treatment within urbanised cells only. Based on previously mentioned assumptions of land-use type and density in section 2.1.2, this amounts to a total dry-proofing area of 22.5% of any given urban cell. Considering the default protection standard of a 2-yr return period, areas of potential application are defined as all inundated urban cells within the 2-yr flood zone that have an inundation depth of less than 1m. In all remaining return periods, dry-proofing is assumed to be applied in inundated areas not excluded by the above delineation.

290   Because this measure is used to protect physical assets within exposed areas (thus reducing monetary damage) but does nothing to alleviate human safety concerns, EAAP will not be altered.

We apply the costs of dry-proofing on a per area basis of buildings within urban cells ($m^2$). The costs, and therefore affordability, are dependent on the income-level of different regions (Hudson, 2020; Aerts, 2018). Thus, we assume a cost of ~$1,300 per square meter for high- and upper-middle-income countries and ~$580 per square meter for lower-middle- and low-income countries. Most existing literature describes in detail the costs posed by dry-proofing within the United States and other developed nations (FEMA, 2021, de Ruig et al., 2020, de Graaf et al., 2013).

## 2.3 Measure evaluation

### 2.3.1 Efficacy

We assess the efficacy of the examined DRR measures by evaluating their ability to reduce (monetary) risk to maintain the *relative-risk constant* target per sub-national region. This efficacy is expressed as the risk reduction achieved by the DRR measure divided by the amount of risk reduction that was required to reach the *relative-risk constant* target. For example, if a DRR measure can achieve the entire amount of required risk reduction to lower future risk levels to the exact level of the risk target, the efficacy of the measure is 1. While it is possible to exceed 1 if a measure reduces risk further than the *relative-risk constant* target, in this study we only report values up to 1. The minimum efficacy is meanwhile assumed to be 0, meaning a measure is unable to reduce any future risk when compared to the scenario in which no action is taken. The efficacy of each measure is calculated for each sub-national region.

### 2.3.2 Benefit-cost analysis

A benefit-cost analysis (BCA) is performed to demonstrate the economic feasibility of a DRR measure, using two indicators, namely Net Present Value (NPV) which is the net return on investment discounted to present value, and the Benefit-Cost ratio (BCR), which is the ratio between discounted benefits and discounted costs. For NPV, the benefits and costs are discounted with a discount rate of 5%. This discount rate is within the range of financial-equivalent rates for adaptation efforts in the face of increasing flood risk (Ward et al., 2017).

Yearly discounted benefits are divided by initial and yearly discounted costs for the BCR. To calculate the benefits of each DRR measure, EAD is calculated for every year of the lifetime of each measure and subtracted from the EAD for every year without the measure. The results of each yearly calculation are summed to determine the total benefits. Costs are calculated by summing investment and capitalised maintenance costs. The costs reported in this paper are in US$2005 at Purchasing Power Parity (PPP) and were adjusted using GDP deflators from the World Bank. As mentioned, indirect benefits such as nature contributions to people (Barbier et al., 2011; Jakovac et al., 2020), are not included in this benefit-cost analysis.

## 3 Results

320    The results of this analysis are presented on several spatial scales, including on the global, regional (i.e., World Bank analytical regions), national, and sub-national level. The results of RCP 6.0 / SSP2 in 2080 are thoroughly examined and visualised in this section, while high- and low-end scenarios (RCP 8.5 / SSP3 and RCP 2.6 / SSP1, respectively) are referred to in order to contextualise the results. The results coming from these three scenarios are available in our study's data repository on the sub-national scale.

### 3.1 Risk estimation assuming no DRR action

325

Future global flood risk values will eclipse those of the present if no action is taken on our coastlines. The need for a no DRR action assessment stems from the theoretical exercise of determining benefits achieved by implementing any given DRR measure. In reality, a future with no DRR action whatsoever is highly improbable. Communities increasingly at risk to coastal flooding will react to the changing conditions. Still, here we quantify this no DRR action scenario as the basis of how much

330    reduction to coastal flood risk is required and is possible. Without any DRR action (i.e., only maintaining the height of current protection structures), EAD is projected to increase by over a factor of 300 and global EAAP is projected to roughly triple by 2080 (Table 1). By the year 2080, we estimate global EAD will be over $1.3 trillion, while global EAAP will exceed 11.5 million people. When considering low- and high-end scenarios, EAD ranges between $650 billion and nearly $2 trillion, while EAAP ranges between 9 million and 18 million. Generally, EAD exhibits an exponential growth pattern while EAAP increases

335    linearly through the examined time-steps of this study. The impacts of current and future coastal flood risk, given at the World Bank Analytical Regional scale in Table 1, are substantially larger in lower-income regions of the world.

**Table 1: Current (2020) and future (2080) EAD (billion USD) and EAAP (millions of people) per World Bank analytical region. The stand-alone future EAD and EAAP values are provided assuming RCP 6.0 / SSP2, while the ranges for future EAD and EAAP are provided across all RCP / SSP combinations, specifically the aforementioned as well as RCP 2.6 / SSP1 and RCP 8.5 / SSP3.**

| World Bank Analytical Region | 2020 EAD | 2080 EAD | 2080 EAD Range | 2020 EAAP | 2080 EAAP | 2080 EAAP Range |
|---|---|---|---|---|---|---|
| East Asia & Pacific | 1.885 | 515.2 | 294.8 – 693.8 | 1.942 | 4.735 | 3.704 – 7.193 |
| Europe & Central Asia | 0.510 | 22.61 | 14.68 – 25.21 | 0.087 | 0.138 | 0.116 – 0.138 |
| Latin America & Caribbean | 0.309 | 17.83 | 14.94 – 19.80 | 0.057 | 0.119 | 0.084 – 0.237 |
| Middle East & North Africa | 0.130 | 33.07 | 18.62 – 40.21 | 0.080 | 0.243 | 0.206 – 0.356 |
| North America | 0.117 | 34.31 | 24.93 – 34.31 | 0.039 | 0.108 | 0.064 – 0.109 |
| South Asia | 0.179 | 187.7 | 89.45 – 297.5 | 0.979 | 3.864 | 3.024 – 6.509 |
| Sub-Saharan Africa | 0.938 | 542.8 | 195.9 – 833.7 | 0.444 | 2.356 | 1.850 – 3.571 |
| GLOBAL TOTAL | 4.068 | 1354 | 653.4 - 1941 | 3.628 | 11.56 | 9.112 – 18.05 |

Between 2020 and 2080, all regions are projected to experience substantial percentage increases in both EAD and EAAP. Due to relatively low existing levels of risk and protection standards, the largest percentage increase to EAD is in South Asia, while the largest percentage increase to EAAP is in Sub-Saharan Africa. The largest share of future risk in absolute monetary terms is found in three regions: East Asia and the Pacific, Sub-Saharan Africa, and South Asia. These regions are projected to experience increases to EAD of roughly $510 million, $540 million, and $190 million, respectively, reflecting the large-scale urbanisation and population growth projected to occur in these regions throughout the remainder of the century.

Of particular concern in these lower-income regions is the magnitude of range, and therefore uncertainty, of potential impacts depending on the RCP / SSP combination considered. For example, we estimate EAD in Sub-Saharan Africa could range between $196 million and $834 million. If one compares a range of this magnitude to that of a higher-income region, such as Europe & Central Asia (with an estimated EAD range of $15 million and $25 million), the economic imperative of comprehensive DRR planning on the global scale is underscored. Beyond monetary concerns, South Asia is projected to see nearly five million people affected by coastal flooding annually by 2080.

Similar patterns can also be seen on the (sub-)national scale. The largest percentage increases to EAD are projected in Bangladesh, Senegal, and Madagascar (three countries with low current levels of risk), while the largest absolute changes are projected in China, India, and Indonesia. The ten countries with the highest future EAAP all belong to the three World Bank regions with highest future EAD mentioned previously. While certain high-income countries, such as The Netherlands, United States of America, and United Kingdom, are also projected to have high future EAAP, lower-middle-income countries represent the largest portion – roughly two-thirds – of future EAAP. Changes in both absolute and relative EAD and EAPP are shown per sub-national region between from current conditions and future conditions in Figure 2.

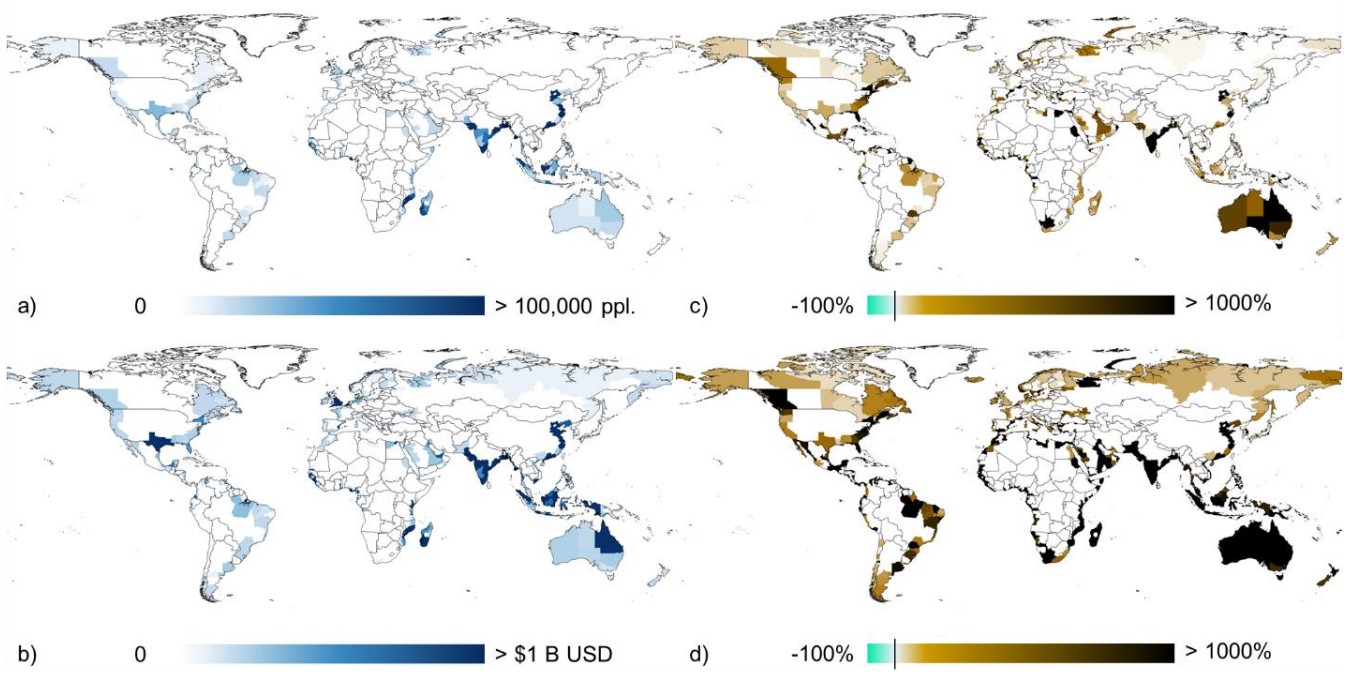

**Figure 2: Global maps depicting simulated sub-national regional changes between current (2020) and future (2080 under RCP 6.0 / SSP2) in: a) absolute EAAP, b) absolute EAD, c) relative EAAP, and d) relative EAD, assuming there are no increases to current protection standards.**

The differences in simulated risk between regions, countries, and sub-national regions result from differences in (changes) in hazard and exposure, as well as the assumed current protection standard per sub-national region. Within our modelling framework, many sub-national regions in South Asia, East Asia and the Pacific, and Sub-Saharan Africa are assigned low current levels of protection from flooding. Meanwhile, the highest estimated current protection standards are found in high income countries. For example, Singapore and The Netherlands are evaluated to have current protection standards equivalent to a 1000-yr return period event. The assumed current flood protection standards have a major influence on the calculated risk,

370    as demonstrated by Ward et al. (2013), which indicates that the level of risk is to a large part influenced by actions taken to prepare for and mitigate flood risk. These global and regional results show that, while spatially localised, coastal flooding has a major economic and human impact on all segments of global society which require DRR action to avoid future impacts.

## 3.2 Future risk target

We estimate that increases to current protection standards are required in most sub-national regions to achieve respective
375    *relative-risk constant* targets. Sub-national protection standards in 2080 if no changes are made to current protection standards are shown in Figure 3, while sub-national future protection standards required to counteract the downgrading of current protection standards and achieve the *relative-risk constant* targets are shown in Figure 4.

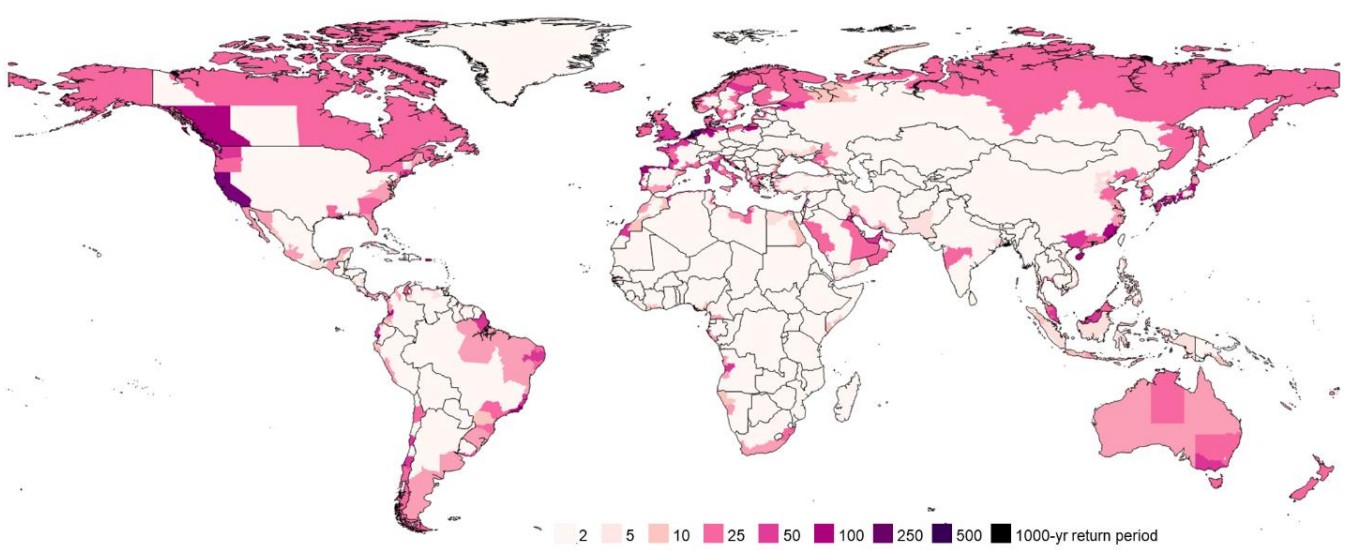

380    **Figure 3: Sub-national protection standards estimated for the year 2080 assuming no changes to current protection standards are made in response to sea-level rise or subsidence.**

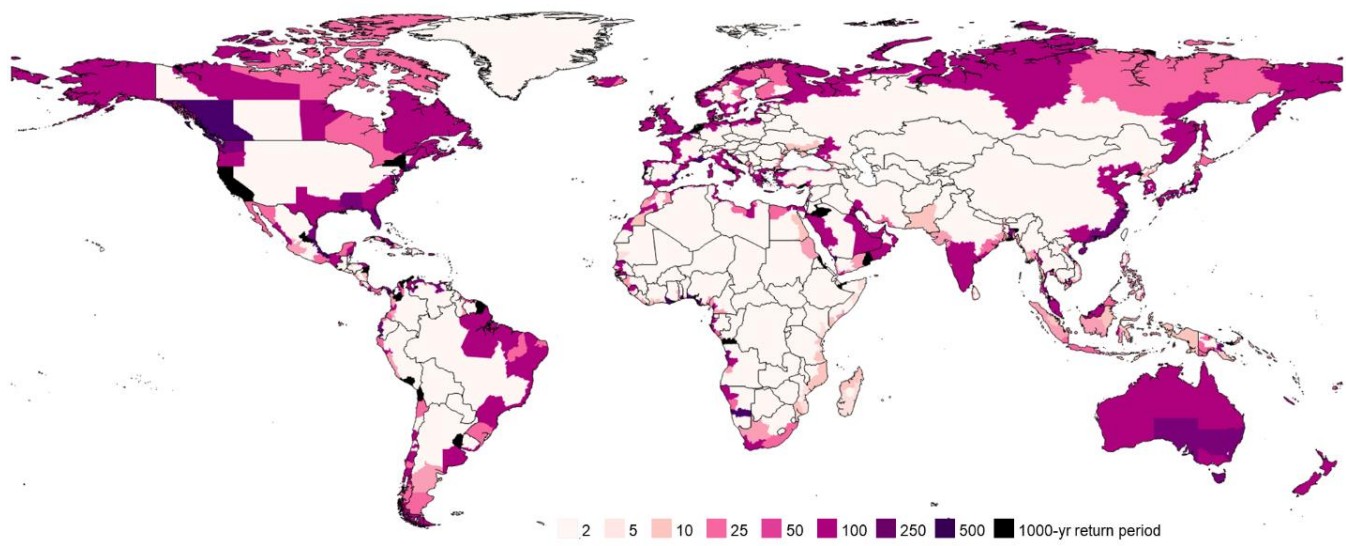

**Figure 4: Estimated protection standards required for sub-national regions to achieve their *relative-risk constant* risk reduction targets in 2080.**

385  As discussed in section 2.1.4, unaltered 2080 protection standards are in most cases lower than 2020 protection standards due to sea-level rise and subsidence. The largest decreases to current protection standards occur in sub-national regions within Portugal, Japan, Brunei, the United States of America, and The Netherlands. All these sub-national regions have current simulated protection standards to protect from an inundation event well above that of the 100-yr return period. By 2080, with increasing hazard, over 230 sub-national regions around the world will have their protection standards minimised to a 2-yr

390  protection standard – the lowest possible level in our analysis.

In some cases, areas with minimal or no existing protection standards against coastal flooding would require significant increases to protection standards, for example by doubling or tripling the order of magnitude of protection provided (e.g., from a 5-yr protection standard to a 500-yr). Many of these instances are found in Sub-Saharan Africa, where many regions have low current protection standards in the modelling framework. Other areas begin with low current flood protection standards,

395  but only require minor increases to reach their target (e.g., from a 3-yr protection standard to a 5-yr). Many of these cases are found in Latin and Central America, where there is topographic relief from the coasts towards the inland. In other regions, relatively small changes in flood protection standards are required because the current protection standards are already high, meaning that actions have already been taken to mitigate flood risk in the past. Existing high protection standards that still require large increases are found in sub-national regions worldwide, with many of these cases located in North America,

400  Europe and Central Asia, and East Asia and the Pacific.

Fifty-two sub-national regions do not require an increase to their respective protection standard in 2080 to reach the *relative-risk constant* target. This can occur for several reasons. Firstly, in some regions future EAD does not increase considerably in the face of changing hazard, due to limited existing exposure or limited simulated change in future exposure, such as along parts of the steep Chilean and Peruvian coasts. Secondly, in some regions localised sea-level rise is marginal, such as in Scandinavia and Russia where post-glacial rebound is present. Thirdly, where large changes in hazard and exposure are projected to occur, the projected overall growth in regional GDP may be higher than the overall growth in EAD, such as in several coastal provinces of China.

## 3.3 Risk reduction provided by DRR measures

In this section, we assess to what extent the *relative-risk constant* targets can be achieved by taking one of the four DRR measures introduced in section 2.2 and applying them individually on the global scale. The globally aggregated findings of the benefit-cost analysis as well as efficacy of each individual measure are summarised in Table 2. While it is assumed that dykes and coastal levees can be built anywhere in the world, the other DRR measures are restricted in terms of suitability. Based on our assessment, foreshore vegetation, zoning restrictions, and dry-proofing are respectively only applicable in 23%, 84%, 45% of sub-national regions evaluated with (future) coastal flood risk due to spatial limitations of these DRR measures as mentioned in section 2.2. The metrics for each DRR measure in Table 2 are calculated only for the applicable regions. Low- and high-end scenario ranges for BCR, NPV, and efficacy are also included for context.

**Table 2: The globally aggregated benefits and costs (billion USD), BCR, NPV (billion USD), and efficacy (\*in applicable regions only) of each individual DRR measure in 2080 under RCP 6.0 / SSP2. Included in parentheses under each DRR measure are the total number of applicable sub-national regions where each DRR measure is assessed, while under BCR, NPV, and Efficacy are the low- and high-end scenario ranges of these values.**

|  | Benefits | Costs | BCR | NPV | Efficacy* |
|---|---|---|---|---|---|
| Dykes and coastal levees (927 regions) | 493 | 86.01 | 7.44 (11.2 – 4.65) | 14.03 (21.05 – 8.769) | 98% (99% - 96%) |
| Foreshore vegetation (227 regions) | 17.2 | 0.366 | 59.8 (72.4 – 32.9) | 0.556 (1.82 – 0.278) | 6% (10% - 3%) |
| Zoning restrictions (784 regions) | 46.5 | 0.027 | 598 (604 – 578) | 0.401 (0.405 – 0.309) | 11% (12% - 8%) |
| Dry-proofing (417 regions) | 242 | 151.5 | 2.10 (2.52 – 1.62) | 4.165 (4.23 – 4.060) | 49% (59% - 38%) |

In the future and under conditions of RCP 6.0 / SSP2, the largest overall benefit of the evaluated DRR measure is for dykes and coastal levees, while the most costly is dry-proofing. The latter results in the lowest overall BCR of any DRR measure.

The benefits of foreshore vegetation and zoning restrictions are smallest in magnitude, while the least costly measure overall is zoning restrictions. Because of the low costs, zoning restrictions achieve the highest simulated BCR of any of the evaluated measures. Still, all four DRR measures achieve globally aggregated BCRs > 1, meaning the long-term benefits of each DRR measure outweigh the costs on the global scale. All globally aggregated NPVs are also positive, with the highest exhibited by dykes and coastal levees, followed by dry-proofing; however, NPV varies greatly for dykes and coastal levees as costs of heightening these structural features outpaces the benefits of doing so under high-end scenario. In general, BCRs, NPVs, and efficacy scores for the DRR measures increase under the low-end scenario (RCP 2.6 / SSP1), and decrease under the high-end scenario (RCP 8.5 / SSP3).

In all World Bank analytical regions, dykes and coastal levees have the highest efficacy in terms of the percentage of the *relative-risk constant* target that they can achieve. Still, considerable reductions are also possible within certain (sub-national) regions by employing foreshore vegetation, zoning restrictions, or dry-proofing. North America exhibits the highest efficacy in achieving its risk target when using dry-proofing (98%), while the Middle East and North Africa achieves high efficacy scores for foreshore vegetation (23%) and for zoning restrictions (34%). South Asia achieves the highest simulated BCRs for all DRR measures of all global regions analysed. Relatively high BCRs are also seen in Sub-Saharan Africa and East Asia and the Pacific. Latin America and the Caribbean achieves relatively low BCRs due to the limited benefits achieved from employing the DRR measures there.

Sub-national values of each DRR measure's efficacy in achieving the *relative-risk constant* targets and BCRs are displayed in Figures 5 and 6, respectively. Over 90% of all sub-national regions can meet their respective *relative-risk constant* targets using at least one of the measures. Of these, approximately 160 regions – or one-fifth – can do so by employing a non-structural DRR measure. To gain further context regarding these DRR measures and their ability to modulate future coastal flood risk, the following subsections examine each measure individually, referring to the relative panels in the following figures.

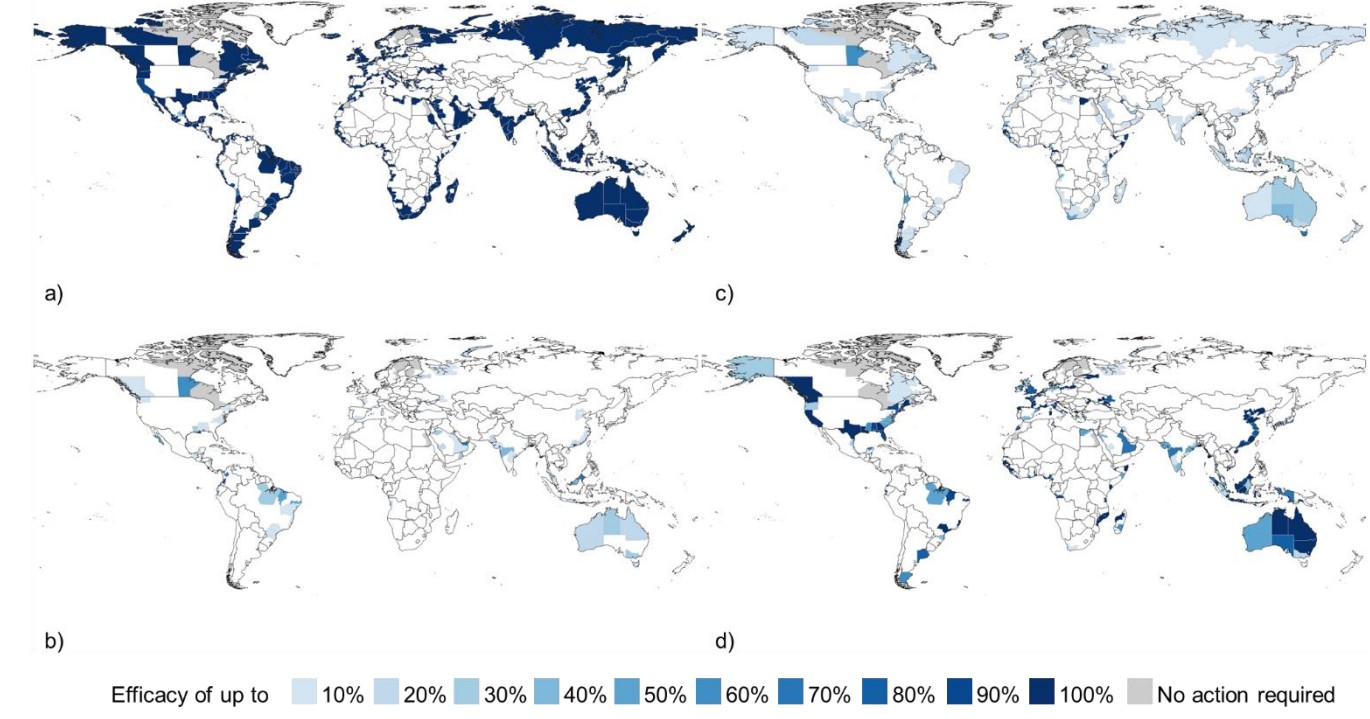

445

**Figure 5: Efficacy in achieving the *relative-risk constant* targets for a) dykes and coastal levees, b) foreshore vegetation, c) zoning restrictions, and d) dry-proofing on the sub-national scale, where light blue indicates lower efficacy and dark blue higher efficacy. Grey areas are sub-national regions where action is not required to achieve the *relative-risk reduction* targets.**

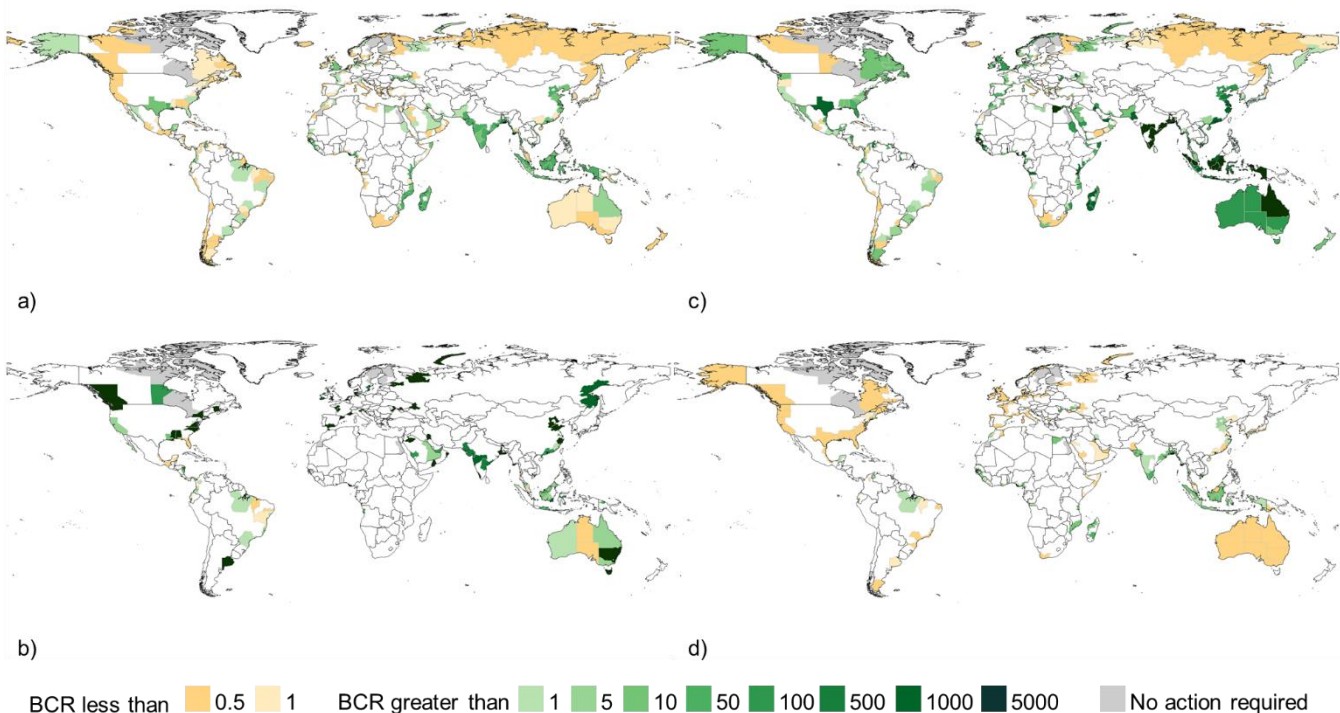

BCR less than ▢ 0.5 ▢ 1    BCR greater than ▢ 1 ▢ 5 ▢ 10 ▢ 50 ▢ 100 ▢ 500 ▢ 1000 ▢ 5000    ▢ No action required

450 **Figure 6: Benefit-cost ratio values for a) dykes and coastal levees, b) foreshore vegetation, c) zoning restrictions, and d) dry-proofing on the sub-national scale, where yellow indicates overall direct costs outweighing overall benefits and green overall benefits outweighing overall direct costs. Grey areas are sub-national regions where action is not required to achieve the *relative-risk reduction* targets.**

### 3.3.1 Dykes and coastal levees

Overall, the results show that implementing dykes and coastal levees has a globally aggregated efficacy of 98% towards achieving *relative-risk constant* targets and global benefits of over $493 billion (Table 2); however, in 10% of sub-national regions estimated to face future flood risk the *relative-risk constant* target is not achieved. A considerable share of the monetary benefits, namely $421 billion, can be achieved in South Asia and East Asia and the Pacific. Globally, the cost of implementing dykes and coastal levees to meet the *relative-risk constant* targets sums to just over $86 billion. The slightly lower efficacy for dykes and coastal levees observed in regions like Europe and Central Asia can be explained by high existing protection standards that require increases above the 1000-yr return period.

Using this DRR measure, almost every sub-national region in the world can achieve their *relative-risk constant* target (Figure 5a), with some exceptions. For Singapore and The Netherlands, protection standards exceeding their current levels would be required to achieve their targets. These two countries are both assumed to have existing protection against a 1000-yr return period flood. Since this level of protection is also the maximum cap within this modelling framework, the simulated residual risk remains above the target. Certain portions of Northern Africa, where coasts are largely unprotected, and the area surrounding the Gulf of Mexico, where substantial amounts of subsidence and sea-level rise are projected to occur (thus degrading existing protection standards), would also require a protection standard exceeding 1000 years. It may be possible to build structural measures that provide a protection standard greater than 1000 years, as is indeed already the case in The Netherlands (Kabat et al., 2009).

Despite high costs, in many sub-national regions the benefits outweigh the costs (Figure 6a). Only in specific sub-national regions, generally those that are sparsely populated or with high projected sea-level rise, do costs outweigh the benefits. The benefits of building new and heightening existing dykes and coastal levees most strongly outweigh the costs of doing so in sub-national areas that have high amounts of current development, and especially those with only limited projected sea-level rise. Likewise, the lowest BCRs are found in lower-income countries with moderate- to low-levels of coastal development. Interestingly, the capital regions of Papua New Guinea, Guinea-Bissau, Cuba, and Suriname all have BCRs < 1 and are unable to reach their risk reduction targets by 2080 using solely dykes and coastal levees. This may be explained by the fact that these growing regions have high levels of concentrated, low-wealth urbanisation which is exposed to high levels of sea-level rise.

### 3.3.2 Foreshore vegetation

The globally aggregated efficacy of foreshore vegetation in achieving *relative-risk constant* targets is roughly 6%, and global reductions to EAD total to more than $17.2 billion (Table 2). The largest reductions are seen in South Asia, East Asia and the Pacific, and Sub-Saharan Africa. Of the $12.4 billion in EAD reduction that is achieved via mangrove restoration, $8.7 billion is found in India, Vietnam, and Nigeria. Conserving salt marshes, meanwhile, shows the most potential in China, representing

485 just over $4.6 billion of the $4.8 billion in reductions to global EAD. In the case of salt marshes, many of these ecosystems have been degraded or lost completely, meaning current marshes may not provide protection for the urban setting.

Although certain salt marsh ecosystems, such as those in northern China and the eastern seaboard of the United States, remain in the vicinity of urbanised areas, the global economic benefits of salt marshes are much lower than those of mangroves. Seeing the larger magnitude of benefits achieved by restoring mangroves as compared to conserving existing salt marshes, it can be

490 suggested that higher BCRs can be achieved using foreshore vegetation as a DRR measure if these areas are not only maintained and conserved, but actively cultivated beyond current extents.

Foreshore vegetation is simulated to reduce risk in one-quarter of all subnational regions with (future) coastal flood risk. Twelve sub-national regions meet or exceed 100% of their risk targets. The average efficacy seen in the global sub-national regions evaluated as having foreshore vegetation is 20%, while the median is 5.7% (Figure 5b). The sub-national regions which

495 do not see reductions as a result of foreshore vegetation are either fully urbanised with minimal potential to support foreshore vegetation, located in unfavourable climate regions for foreshore vegetation, or have favourable conditions to support foreshore vegetation but not near any flood-prone populations

The global costs of foreshore vegetation are estimated to be $366 million. Sub-national regions within China, Vietnam, and the Philippines all possess among the highest BCRs, with similarly high BCRs in upper-latitude sub-national regions in Canada

500 and Russia. The lowest BCRs are found in several countries within Latin America and the Caribbean and Sub-Saharan Africa (Figure 6b). This is likely because many major mangrove areas within this part of the world are not around major population centres, but rather on small islands and sparsely populated coastlines.

### 3.3.3 Zoning restrictions

Zoning restrictions achieve EAD reductions of $46 billion by 2080 worldwide (Table 2). Generally, exposure reduction as a

505 single solution is only effective in certain contexts, namely in areas where current coastal urbanisation is limited but future urbanisation is widely projected. Just over half of the risk reductions are realised in East Asia and the Pacific, the region of the world that shows the highest projected levels of population growth through the 21[st] century. The Middle East and North Africa, Sub-Saharan Africa, and South Asia also experience large risk reductions through zoning restrictions. Meanwhile, Latin America and the Caribbean experiences limited risk reduction via this measure due to limited coastal regions where additional

510 urbanisation can occur, even though moderate amounts of growth are projected. Meanwhile, North America as well as Europe and Central Asia show low potential because their coastlines are already highly urbanised, and therefore the zoning measures are not impactful as implemented in the model. Notably, three countries count for almost two-thirds of total global risk reduction through this DRR measure: India, China, and Indonesia. Other countries that do not experience large risk reduction via other DRR measures, such as Guinea, Senegal, and Cameroon, do show high efficacy towards achieving their *relative-risk*

*constant* targets through zoning restrictions. Japan and Australia are the high-income countries that experience the largest risk reductions, with $423 million and $373 million fewer damages, respectively, expected by 2080.

The globally aggregated efficacy of zoning restrictions towards achieving *relative-risk constant* targets is approximately 11%. Sub-national regions within 20 countries, mainly low-income, but also in countries such as Japan and Spain, can achieve their respective risk reduction targets by limiting future exposure in the 1000-yr floodplain. One-fifth of sub-national regions
achieve 20% or more of their *relative-risk constant* targets (Figure 5c). Zoning restrictions, as modelled in this research framework, show the highest efficacy towards achieving *relative-risk constant* targets in sub-national regions within low-income countries. Often, these countries are projected to urbanise substantially along the coasts.

While the benefits of zoning restrictions are less than one-tenth of those provided by dykes and coastal levees globally, this DRR measure is highly cost effective. The minimal costs associated with this DRR measure mean that, globally, zoning
restrictions achieve the highest BCR of any modelled measure (Figure 6c). The cost-effective nature of this measure is compounded in certain global regions because low-income countries within these regions have both the highest levels of projected population growth (which can be spatially altered by this DRR measure) as well as lower costs of implementation when compared to high-income countries. Costs run the highest in Europe and Central Asia for zoning restrictions because of the high-income nature of its countries as well as the large amount of sub-national regions within these countries, each requiring
their own zoning policy development and administration. As mentioned, for our analysis we assume these administrative costs are the only direct costs of this DRR measure. The limitations of this specific assumption are discussed further in section 3.4.

### 3.3.4 Dry-proofing

Dry-proofing can be highly effective in reducing EAD, with a global reduction to EAD by as much as $242 billion (Table 2), or nearly half of the reductions needed to achieve the set *relative-risk constant* targets. While half of this risk reduction is
found in East Asia and the Pacific ($158 billion), the benefits of dry-proofing are generally spread across all global regions. The smallest risk reductions are seen in Latin America and the Caribbean (under $2 billion), with relatively few major urban centres along vulnerable coastlines when compared to the eastern hemisphere, the exception being the (east) coast of Brazil. Europe and Central Asia also exhibit lower reductions when compared to other global regions; here, existing high protection standards limit the impact of a DRR measure that mainly takes aim at lower return period events.

The countries that are projected to experience the highest reductions are China (with the Shanghai direct-administered municipality alone accounting for one-third of total countrywide reductions), Vietnam, Indonesia, and India, each with reductions in the order of magnitude of tens-of-billions. With roughly $4.5 billion in dry-proofing reductions, the high-income country with the largest amount of damage reductions to be achieved via dry-proofing is the United States of America, with a large portion of these reductions found in the states of New York and Texas. This DRR measure can meet *relative-risk constant*

targets in sub-national regions within 46 countries of varying income levels and urbanisation patterns, ranging from Singapore and France to Yemen and Somalia (Figure 5d). While dry-proofing is not applicable in all contexts within this modelling framework (e.g., not modelled for rural settings or in semi-urban or peri-urban settlements, which are prevalent in certain global contexts), roughly two-thirds of the coastal sub-national regions where dry-proofing is applicable achieve at least 50% their *relative-risk constant* targets. These results show that dry-proofing is a measure that is most effective in highly urbanised

coastal settings with low projected future inundation depths (of less than 1m).

Sub-national regions with existing patterns of intensive urbanisation along coastlines, for example the eastern seaboard of the United States of America and eastern China, show the highest potential for risk reduction with this DRR measure. Coincidentally, the highest costs for this measure are also found in these areas, with dry-proofing within the United States of America alone costing an estimated $53 billion and China $33 billion, leading to low BCR values (Figure 6d). In fact, in the

United States of America, only New Hampshire scores a BCR > 1. Total global costs sum to $151 billion, meaning that while benefits still outweigh costs, the return-on-investment of this measure is smaller than other DRR measures. This, combined with the fact that this measure is a solution employed by individuals and not on larger scales, could make a dry-proofing strategy a challenging one for much of the world. Although they observe high levels of risk reduction, low BCRs in North America and Europe and Central Asia mean that a decision maker may select a different DRR measure. BCRs are largest in

Sub-Saharan Africa and South Asia, particularly in India and Indonesia, both of which have several major urban centres on their coastlines.

### 3.4 Uncertainty, limitations and recommendations for future research

Uncertainty in our analysis originates from several sources, including data inputs and modelling assumptions. This is also discussed in other global-scale coastal flood risk assessment literature. With regard to scenario uncertainty, Rohmer et al.

(2021) state that adaptation costs are most sensitive to RCP used, while EAD is more sensitive to SSP. RCPs ultimately drive sea-level rise projections, which are also based on thermal expansion, global surface air temperature and ocean dynamic sea level from the Coupled Model Intercomparison Project 5 (CMIP5) with IPCC AR5 estimations of ice and land water contributions complemented with the re-evaluation of Antarctic contribution from SROCC. These uncertainties are combined based on the probabilistic model described in Le Bars (2018). While here we mainly explore the results of RCP 6.0 / SSP2,

the results of RCP 2.6 / SPP1 and RCP 8.5 / SSP3 show that both flood risk estimations and DRR measure performance are impacted by the selected combination of RCP / SSP.

Similar patterns of sensitivity are seen in complementary studies. Tiggeloven et al. (2020) sees the largest sensitivity for global adaptation costs stemming from sea-level rise. Indeed, the largest source of scenario uncertainty, according to Hinkel et al. (2021), relates to future coastal adaptation scenarios, which can influence future coastal flood risk by factors of 20.0–26.7. It

is this exact source uncertainty that we explore with our analysis by employing several DRR measures, reaffirming that future

coastal flood risk depends greatly on which action is taken by decision makers (Hinkel et al., 2014). An uncertainty framework for coastal hazard assessment, as developed by Stephens et al. (2017), could be used to overcome these and other sources of uncertainty such as data input uncertainty (e.g., DEM and exposure maps); however, this sort of framework is designed to guide local assessments and has not yet been expanded to the regional and global scales.

In general, we acknowledge that the assumptions used in our global analysis do not capture a fully representative picture of what the modelled DRR measures would be in reality, especially in terms of their actual effectiveness (as opposed to their efficacy), variations around the world, and potentially dynamic nature. Several assumptions are made specifically in the implementation of our DRR measures. For example, we assume the percentage of occupancy type per grid cell to be the same for all locations, whilst in reality it is spatially heterogeneous. We also assume building density per occupancy type. An

improvement to our analysis could be made by using machine learning to improve accuracy of urban land cover and building types (Hecht et al., 2015; Huang et al., 2018). While we have assumed a rapid adoption of DRR measures and full efficacy/uptake, timing and rate of a commitment to adaptation varies per country (Haasnoot et al., 2021), which we do not consider here. Moreover, we have assumed that the DRR measures do not experience any failure below the threshold of provided protection standards; in reality, violent storm events could partially damage or destroy the DRR measures, in

particular sensitive ones such as foreshore vegetation. Meanwhile, coastal squeeze caused by sea-level rise and urbanisation could cause further reductions of effectiveness of certain DRR measures (Torio & Chmura, 2013). These limitations could lead to underestimation of costs of (re)implementation and overestimation of benefits provided by measures if they were to experience such shortcomings and failures.

The direct benefits provided by the modelled DRR measures are determined in a similar fashion for each measure. The direct
costs associated with these measures, however, are determined in fundamentally different manners (as described in the subsections of 2.2) out of necessity. With certain DRR measures there are additional costs which could also considered; we do not include these indirect costs in this analysis. For instance, infrastructure disruptions might occur more frequently if regions rely on dry-proofing instead of dykes and coastal levees, while zoning restrictions could also prevent infrastructure disruptions (Koks et al., 2022; Nirandjan et al., 2022). Other examples of indirect costs that are not included in this analysis but could

provide additional value to the field of risk assessment include the cost of ecosystem disruptions with structural measures, opportunity costs associated with not developing in flood-prone areas, and so on. Furthermore, the nature contributions to humans of the DRR measures are not included in our analysis, resulting in an underestimation of benefits provided by certain DRR measures. Any one of these indirect costs or benefits are strongly recommended as a future avenue of research.

We also recommend the continued expansion of modelled DRR measures within the same risk assessment framework to allow
for a more comprehensive view of coastal flood risk. Candidates for incorporation into global flood risk modelling frameworks could include other nature-based solutions, for example salt marsh restoration and coral reef restoration, as well as managed

retreat. Beck et al. (2022) did recently assess the potential for coral restoration; as with the foreshore vegetation modelled in this study, the measure may have limited spatial applicability on the global scale but could have high levels of impact on small island developing nations and other settings highly vulnerable to climate change. Managed retreat, meanwhile, in theory can

be applied in some locations around the world but may be difficult to implement in a practical sense. The modelling of this DRR measure could involve agent-based modelling based on locally surveyed information to represent the associated dynamics, as performed on the local scale by Tierolf et al. (2023).

By employing any of the modelled DRR measures individually, we find that 84 sub-national regions cannot achieve their respective *relative-risk constant* targets by 2080. While here we investigated the measures individually, the actual

implementation of a DRR strategy will often consist of executing several DRR measures at once, or in a hybrid form (Dedekorkut-Howes et al., 2020; Dissanayaka et al., 2021; Lawrence et al. 2019). Indeed, DRR measure hybridisation could help increase the DRR potential in many of these 84 sub-national regions while minimising costs (Sutton-Grier et al., 2015, Tiggeloven et al., 2022). Future research could further investigate DRR measures hybridisation, as has been the case already with combining zoning restrictions and dry-proofing (Richert et al., 2019), or mangroves and dykes (Tiggeloven et al., 2022).

If DRR measure hybridisation is to be explored further, additional information on nature contributions to people, indirect costs, and unintended side effects (such as increased urbanisation in safer areas (Haer et al., 2020) and others) would be required to create a more comprehensive and holistic view of coastal flood risk reduction in the future. Building upon the methodology presented in this study, additional research on any of these facets can provide insights into such complexities.

## 4 Conclusions

We project immense increases to coastal flood risk in coming decades if no action is taken to limit the growth of disaster risk; expected annual damages are projected to exceed $1.3 trillion by 2080, and under certain climatic and population conditions could reach $2 trillion. Without action, leaders and decision makers in all countries, regardless of income level, will see their societies negatively impacted. Disaster risk reduction is needed to prevent or reduce these impacts. We show how different DRR measures can be implemented to reach a certain risk reduction (i.e., risk reduction targets set by the *relative-risk constant*

objective). We show that, globally, dykes and coastal levees achieve $493 billion in risk reductions, dry-proofing $242 billion, zoning restrictions $46.5 billion, and foreshore vegetation $17.2 billion. The globally-aggregated efficacy of these four DRR measures meanwhile, is 98%, 49%, 11%, and 6%, respectively. While the global aggregates of efficacy and BCRs favour dykes and coastal levees, regionally and sub-nationally we show that the modelled alternatives can yield similar levels of efficacy and higher BCRs.

Indeed, the on-the-ground design and implementation of DRR measures requires site-specific and detailed local information. A more detailed and accurate depiction of global DRR measure implementation could potentially be achieved as a result. This

is not the aim of our research endeavour. Here, we have used a globally applicable model to produce proxy results for DRR measure implementation, which (in data-scarce regions) could allow end-users such as UN-affiliated organisations, the World Bank, and (inter)national adaptation strategists to prioritise actions and further investigation.

As illustrated by our work, 90% of all sub-national regions in the world can achieve their respective *relative-risk constant* targets by using at least one of the modelled DRR measures. Over 160 of these sub-national regions – roughly one-fifth – can achieve their *relative-risk constant* targets via a non-structural DRR measure. This suggests to decision makers in these areas that alternatives to traditional coastal flood risk reduction could be embraced while often lowering overall direct costs. For 84 sub-national regions, though, none of the DRR measures investigated here meet the *relative-risk constant* target, meaning that

something more must be done in these areas, such as exploring other DRR measures and/or the hybridisation of different DRR measures.

## 5 Code availability

The code used to perform this analysis is available upon request.

## 6 Data availability

The input data for hazard and exposure values were created specifically for this analysis by project partners who have generously agreed to making these valuable resources available for public use. The vulnerability values are taken from an open-source publication. Full results of our analysis, both those presented thoroughly in the main text as well as those mentioned briefly, are also available for public use.

Data can be retrieved from this repository: https://zenodo.org/records/10637089

## 7 Author contribution

EM, TT, TH, and PW expanded the existing GLOFRIS framework to include new modules that allowed for the modelling of alternative DRR measures. EM conducted the main analysis, with guidance from TT in operationality of GLOFRIS. DLB, SM, DE, and FSW contributed to the development of hazard data. BB, AB, and WL contributed to the development of exposure

data. TH and PW assisted EM in the framing of the analysis for this written text. All authors reviewed the text and provided suggestions for revisions to the final manuscript as presented here.

## 8 Competing interests

One of the co-authors listed is a member of the NHESS editorial board.

## 9 Acknowledgements

The research leading to these results received funding from The Netherlands Organisation for Scientific Research (NWO) in the form of a VIDI grant (grant no. 016.161.324) and the Future Water Challenges II project, funded by The Netherlands Ministry of Infrastructure and Water Management (PBL). The computational work was largely carried out on the Dutch national e-infrastructure with the support of the SURF Cooperative, to whom much gratitude is given.

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
