# Peer review of "The potential of global coastal flood risk reduction using various DRR measures"

_Natural Hazards and Earth System Sciences, 2022_

## Author Comment (AC1)

**REFEREE 1**

**Comment 1: The manuscript by Mortensen et al. employs a global flood model to estimate the effectiveness of a range of disaster risk reduction (DRR) measures in limiting future flood risk, focusing on potential direct impacts. In their assessment, the authors use the constant relative-risk objective, with respect to the regional Gross Domestic Product (GDP). The paper is well structured, clearly written, the results are presented in detail and the limitations are acknowledged and outlined in a clear manner. The study produces interesting results regarding the effectiveness of the different measures in reducing risk and regarding the cost-benefit ration of these measures and are therefore interesting for developing regional strategies to manage coastal flood risk.**

**I would recommend the study for publication – nevertheless, I would like to post some comments that I believe need to be addressed or may be useful for the authors to improve the paper:**

Response 1: We thank the referee for their assessment of our manuscript, and appreciate the general feedback given for improvement. We specifically reply to the points raised below.

**Comment 2: I am unsure why the authors refer to the measures that they are exploring as DRR measures. I do understand that floods can be disastrous but, if I am not mistaken, the authors are not assessing risk based only on high-impact low-probability events that could lead to disasters; rather, they estimate flood probabilities integrating over a range of return periods (ranging from events with 2-year return period, which hardly constitute disasters, to events with 1000-year return period). To my knowledge, these measures are usually referred to in the literature as coastal adaptation measures or grouped under the IPCC coastal adaptation typology categories. I find this potentially confusing and would suggest the authors to either explain clearly why the term DRR is used or refer to the IPCC terminology for coastal adaptation.**

Response 2: The authors believe that coastal adaptation, disaster risk reduction, and climate change adaptation are all interrelated terms. We will include the following text in the second paragraph of section 1 of the manuscript to clarify our decision to use disaster risk reduction:

> "Forward looking disaster risk reduction (i.e. prospective disaster risk management) that examines potential future risks under climate change scenarios specifically also examines future risk, as does climate adaptation. Indeed, there is a call in the policy and science literature to bridge the silos between these domains (UNDRR, 2020). In our analysis, because the measures as implemented to protect against a quantifiable return period of inundation

in the future, we use the term DRR to refer to any actions taken to address changes in coastal flood risk."

An extra point of point of clarification in response to the referee – our framework does not only assess impacts from large disasters. We calculate impacts for return periods between 2 years (high frequency) and 1000 years (low frequency) and calculate risk by integrating across impacts for all return periods.

**Comment 3: The authors use a GIS-based inundation model, considering attenuation of water levels and, if I am not mistaken, waves. It is however not clear to me how waves have been accounted for in the total water level. Could the authors please clarify? Also, I would assume that the wave values that the authors are using refer to offshore waves; or does the model propagate waves to the near shore? (and how near is the "near shore", since wave height will change considerably as waves approach the coast)**

Response 3: The hazard dataset that we have used in our analysis does not include waves or wind-related surface interactions; a simple bathtub model with dynamic resistance is used in our modelling scheme. Instead, we account for wave height in the implementation and performance of DRR measures that are hazard specific, namely dykes and levees and foreshore vegetation. For dyke height increases, current dyke heights are derived for coastal segments and perpendicular coast-normal transects (766,034 transects in total). For each transect, bed levels are constructed and, subsequently, hydrodynamic conditions and near-shore wave attenuation are derived. Lastly, the resulting sea water levels are translated into dike heights. For further details of this methodology, refer to Tiggeloven et al. (2020) and van Zelst et al. (2020). A similar methodology is used to account for wave height reduction resulting from foreshore vegetation, with further details found in Tiggeloven et al. (2022). We will revise the final paragraph of section 2.1.1 of the manuscript as follows:

> "We follow the Peak Over Threshold (POT) method and we fit the Generalized Pareto Distribution (GPD) on the peaks that exceed the 99th percentile surge level. From there we derive estimated sea levels for various return periods. These computed sea levels are then used as input for a GIS-based inundation model using the MERIT DEM (Yamazaki et al., 2018), as described in Tiggeloven et al. (2020), to simulate the inundation. This is a static flood model that simplifies all dynamic processes into a single attenuation factor of the water levels over land (Vafeidis et al., 2019), resulting in a simple bathtub model with static forcings instead of a more complex dynamic inundation model framework. The flood maps do not include wind-waves or future changes in waves and storminess. Rather, (nearshore) waves are accounted for in calculation of the hazard-specific DRR measures effect on inundation levels, discussed further in sections 2.2.1 and 2.2.2 below.

Results of RCP6.0, an intermediate climate change scenario (O'Neill et al., 2017), are explored here in the main text. Additional RCPs are available in the supplement."

With this revision to section 2.1.1, the text describing the method of accounting for waves in section 2.2.1 is now more clear. In section 2.2.2 of the manuscript, the following sentences will be added to clarify how waves are accounted for in modelling foreshore vegetation effects on coastal flood hazard reduction:

"Similar to Tiggeloven et al., (2022), here wave conditions are derived from the ERA-Interim (Dee et al., 2011) reanalysis using a peak-over-threshold approach. To determine the wave attenuation over a foreshore and the resulting significant wave height relevant for the flood defense on a transect, we search an existing lookup-table (van Rooijen et al., 2016) of hydrodynamic numerical modelling results for combinations of foreshore slopes, vegetation covers and hydrodynamic conditions (van Zelst et al., 2021). These searched wave heights are modelled at regular intervals along a steady slope, both with and without salt marsh or mangrove vegetation. Wave angle of incidence is assumed coast normal. Wave attenuation along the vegetated coastlines is determined based on the closest match between the derived transects characteristics and look-up table results."

**Comment 4: Although the authors outline very clearly the limitations of the study, there is hardly any discussion on uncertainty and how this is addressed – where do the main uncertainties in the results stem from? I guess it would be too much to ask the authors to conduct an uncertainty analysis but there is substantial literature regarding flood risk assessments and the authors should at least discuss this issue.**

Response 4: We thank the referee for this valid concern. We will incorporate several lines of text describing the uncertainties that might arise from our analysis. The lines will be incorporated into section 3.4, with the title of this section being revised to reflect our addition:

"Uncertainty in our analysis originates from several sources, including data inputs and modelling assumptions. This is also discussed in other global-scale coastal flood risk assessment literature. With regard to scenario uncertainty, Rohmer et al. (2021) state that adaptation costs are most sensitive to RCP used, while EAD is more sensitive to SSP. RCPs ultimately drive sea-level rise projections, which are also based on thermal expansion, global surface air temperature and ocean dynamic sea level from the Coupled Model Intercomparison Project 5 (CMIP5) with IPCC AR5 estimations of ice and land water contributions complemented with the reevaluation of Antarctic contribution from SROCC. The uncertainties are combined based on the probabilistic model described in Le Bars (2018).

Tiggeloven et al. (2020) sees the largest sensitivity for global adaptation costs stemming from sea-level rise. Indeed, the largest source of scenario uncertainty, according to Hinkel et al. (2021), relates to future coastal adaptation scenarios, which can influence future coastal flood risk by factors of 20.0–26.7. It is this exact source uncertainty that we explore with our analysis by employing several DRR measures, reaffirming that future coastal flood risk depends greatly on which action is taken by decision makers (Hinkel et al., 2014). Ultimately, an uncertainty framework for coastal hazard assessment, as developed by Stephens et al. (2017), could be used to overcome these and other sources of uncertainty such as data input uncertainty (e.g., DEM and exposure maps); however, this sort of framework is designed to guide local assessments and has not yet been expanded to the regional and global scales."

**Comment 5: Following my previous point, uncertainty (in e.g. socio-economic development) is often addressed with the use of scenarios. The authors use only one scenario combination (SSP2-RCP6.0), which is a middle of the road scenario; I am unsure what the value of this is since it gives us practically no information about the potential range of uncertainty. In this case, either a second scenario should be used or the authors should rather opt for a high- or low-end scenario which would indicate the upper or lower boundaries. Of course, there is value in comparing the different measures, however, in a different scenario combination results could look very different.**

Response 5: Indeed, as the referee mentions, our primary focus for this analysis was the effect of the DRR measures on future coastal flood risk impacts. In addition to the aforementioned text detailing potential sources of uncertainty, we intend to include certain scenario combinations and subsequent results in the supplement of the manuscript. The combinations SSP2-RCP2.6, SSP2-RCP4.5, and SSP2-RCP8.5 are therefore included in the supplement. The following text will be included in section 3.3 of the manuscript:

> "While here we only present the results of SSP2-RCP6.0, additional results are available for other SSP-RCP combinations in the supplement, namely combinations with varying RCPs. These additional results show that while the overall magnitude of increases to future risk remain substantial regardless of combination, larger EAD values can be expected with higher-end RCPs. Additionally, we see the effectiveness of certain DRR measures, specifically foreshore vegetation and zoning restrictions, decreasing with higher-end RCPs."

Specific value ranges will be added to Table 2 of the manuscript to reflect this textual addition, thus representing the range of potential outcomes under various climate scenarios. In our analysis, we specifically wanted to focus on the impact of climate change on the future of DRR, and not necessarily socioeconomic development or other sources of uncertainty. We chose this route, among other reasons, because of

the large amount of attention currently given on the global stage to concerted efforts in reducing the amount of warming that occurs on Earth due to the impact that this warming has on sea-level rise (Slangen et al., 2022). This notion is seen in IPCC reports and annual COP meetings. Nevertheless, we will include the following sentence in section 3.4 of the manuscript:

> "While here we chose to focus only on the impact of climate change (RCPs) on the future of DRR, further research could integrate different SSP storylines to investigate socio-economic change rather than the effects of climate change as we do in the paper."

**Comment 6: I understand the need for a no-measures assessment. However, I believe that it should be clearly pointed out that this is just a theoretical exercise since, in reality, there will be a response to flooding and adaptation will take place in one form or another, at some point in the century.**

Response 6: We agree with the referee and will add the following sentences in the opening of section 3.1 of the manuscript to clarify this point:

> "The need for a no DRR action assessment stems from the theoretical exercise of determining benefits achieved by implementing any given DRR measure. In reality, a future with no DRR action whatsoever is highly improbable. Communities increasingly at risk to coastal flooding will react to the changing conditions. Still, here we quantify this no DRR action scenario as the basis of how much reduction to coastal flood risk is required and is possible."

**Comment 7: Foreshore vegetation can be partly effective in reducing flood risk – however, a high-end event would destroy foreshore vegetation, thus limiting its protective effects for the years to come. I assume that this has not been considered, I however believe that it would be useful to discuss.**

Response 7: We assume that all measures – foreshore vegetation and otherwise – are implemented and do not experience any failure below the threshold of the protection standard provided. In reality, high-end events could reduce the effectiveness of any risk reduction measure, thus necessitating their repair or replacement.

If we were to include actual benefit losses and subsequent replacement cost calculations within our analysis due to destruction of foreshore vegetation, we would expect overall risk levels would slightly increase during the few years needed for the foreshore vegetation to rejuvenate or be replaced. This could be very interesting to investigate in future work, but here it serves as a limitation as we do not focus on pathway evolution through time, but rather on one point in the future (in our case 2080). Certain studies, such as Haer et al. (2020) and Schlumberger et al. (2022) look at time dynamics of adaptive action, albeit at much smaller scales. This timesensitive component of adaptation was not within the scope of this manuscript, but is interesting to explore further.

We will include this important point raised by the referee as a topic of discussion in section 3.4 of the manuscript with the following text:

> "While here we have assumed our DRR measures do not experience any failure below the threshold of provided protection standards, violent storm events could in reality partially damage or destroy the DRR measures, in particular sensitive ones such as foreshore vegetation. This limitation results in the potential underestimation of costs of (re)implementation and overestimation of benefits provided by measures if they were to experience such failures."

**Comment 8: My last point is a suggestion: based on my experience, many of the differences in global flood impact assessments stem from the calculation of the floodplains. I would personally find it useful if the authors would make their floodplains freely available (not only upon request as this usually does not work) so that others can use them to produce estimates that are comparable. I believe there could be a lot of added value for the research community if everyone conducting global or continental impact assessments made their floodplains openly available.**

Response 8: We agree with the referee's comment on the importance of making data publicly available. In fact, the hazard data used in the most prior application of GLOFRIS is already publicly available via the World Resource Institutes' Aqueduct webtool (www.wri.org/publication/aqueduct-floods-methodology). We intend to make the updated hazard extent maps created explicitly for our analysis available for public use under creative commons licensing.

**Comment 9: I hope my comments help the authors to strengthen this very good manuscript.**

Response 9: We again thank the referee for their time in reviewing our manuscript and for providing useful comments that have improved the study.

**REFEREE 2**

**Comment 10: In this study, the authors estimate the effectiveness of DRR measures for coastal flooding and provide sub-national risk estimates. This is a complex topic given the dynamics in hazard, exposure and vulnerability components. DRR measures are very important for reducing flood risk. Firstly, thank you for addressing this important aspect in flood risk management.**

Response 10: We thank the referee for their assessment of our manuscript, and appreciate the general feedback given for improvement. We specifically reply to the points raised below.

**Comment 11: The authors mention that one of the novel aspects of the study is the global scale of analysis. Unfortunately, I have major concerns regarding the assumptions behind the risk computation and hence, the overall take away from this study.**

Response 11: We indeed assert that our global-scale analysis is novel in that it examines a previously unexplored set of DRR measures all within the same global flood risk model (GFRM), one of several research gaps that was identified by Ward et al. (2015). This article, which discusses the usefulness and limitations of GFRMs, was erroneously excluded from our list of references and has been textually inserted where appropriate, specifically in discussions of uncertainty and limitations of our analysis (see below).

**Comment 12: I like the concept of risk constant. However, many other assumptions are quite vague to generalize. The possibility to implement DRR measures and their effectiveness to reduce risk are very diverse across regions and countries. For example, the assumption such as a constant % of dry-proofed area and urban cell composition are too simplified for a cost analysis and could be wrong for many regions. The same with the generalized costs of zoning. The authors do mention that as a limitation, however, it is a significant limitation that questions the credibility and usability of the results presented.**

Response 12: The referee brings up a fair critique of some of the assumptions used in our global modeling framework. We will add the following text to clearly state these limitations in section 3.4 of the manuscript:

> "Several assumptions are made in the implementation of our DRR measures. For example, we assume the percentage of occupancy type per grid cell to be the same for all locations, whilst in reality it is spatially heterogeneous. We also assume building density per occupancy type. An improvement to our analysis could be made by using machine learning to improve accuracy of urban land cover and building types (Hecht et al., 2015; Huang et al., 2018). Furthermore, while we have assumed a rapid adoption of DRR measures and full

effectiveness/uptake, timing and rate of a commitment to adaptation varies per country (Haasnoot et al., 2021), which we do not consider here."

On the credibility and usability we want to stress the purpose of global analysis versus local analysis, as also outlined in Ward et al. (2015) on the advantages and disadvantages of global scale analysis. We argue that global analyses such as ours support dialogue with stakeholders, including policy and decision makers, and identify priority regions for action. Conducting global-scale risk analysis for disaster risk reduction is essential for gaining a comprehensive understanding of interconnected risks, addressing transboundary challenges, facilitating comparative analysis, promoting collaboration, and developing effective strategies to prevent and mitigate global risks. We intend our framework to be used to highlight potential savings (in the form of expected damage reductions) through strategies which increase DRR at the sub-national scale.

However, as the referee also highlights, when moving towards implementation of individual DRR measures identified by this and other global studies, detailed studies should be performed using local models and data. This sentiment is captured by several other proposed textual additions to the manuscript that are explicitly spelled out in this response (see responses 4, 14, and 15).

**Comment 13: I see that two out of the four DRR measures – dikes and foreshore vegetation are part of the previous work done by the co-authors (as cited in this manuscript). The new findings are the effects of dry-proofing and zoning (please clarify if I am missing something here).**

Response 13: We will add the following text to section 1 of the manuscript to clarify the novelty of our analysis:

> "In our analysis, we have developed and modelled dry-proofing and zoning restrictions as DRR measures, which has never been done before on the global scale. We have also incorporated previously unconsidered costs for foreshore vegetation, namely mangrove restoration costs. To fully compare these new findings, the flood risk impact reduction potential of dykes and levees as well as foreshore vegetation are also recalculated using new hazard and exposure data that were developed explicitly for this analysis."

**Comment 14: I strongly believe that there is a definite need to motivate the implementation of DRR measures. However, the generalized assumptions made in the study without considering local processes make the risk numbers at the Global level questionable. Also, the authors have not provided uncertainty ranges or any sort of validation for any of the reported values (e.g. EAD and EAAP; risk-reduction due to measures).**

Response 14: Thank you for this excellent suggestion to include uncertainty ranges. We will include additional scenarios combinations in the supplement and text within the main body of the manuscript to address scenario uncertainty. In our analysis, we specifically wanted to focus on the impact of climate change on the future of DRR, and not necessarily socioeconomic development or other sources of uncertainty. The combinations SSP2-RCP2.6, SSP2-RCP4.5, and SSP2-RCP8.5 have therefore been included in the supplement. The following text will be inserted in section 3.3 of the manuscript:

> "While here we only present the results of SSP2-RCP6.0, additional results are available for other SSP-RCP combinations in the supplement, namely combinations with varying RCPs. These supplementary results show that while the overall magnitude of increases to future risk remain substantial regardless of the SSP-RCP combination, larger EAD values can be expected with higher-end RCPs. Additionally, we see the effectiveness of certain DRR measures, specifically foreshore vegetation and zoning restrictions, decreasing with higher-end RCPs."

Specific value ranges will be added to Table 2 of the manuscript to reflect this textual addition. To discuss further potential sources of uncertainty, we will add text to section 3.4 of the manuscript:

> "Uncertainty in our analysis originates from several sources, including data inputs and modelling assumptions. This is reflected in other global-scale coastal flood risk assessment literature. With regard to data inputs, Rohmer et al. (2021) state that adaptation costs are most sensitive to RCP used, while EAD is more sensitive to SSP. This notion is supported by Tiggeloven et al. (2020), which sees a majority of the sensitivity for global adaptation costs stemming from sea-level rise. Indeed, the largest source of uncertainty, according to Hinkel et al. (2021), relates to future coastal adaptation scenarios, which can influence future coastal flood risk by factors of 20.0–26.7. It is this exact source uncertainty that we explore with our analysis by employing several DRR measures, reaffirming that future coastal flood risk depends greatly on which action is taken by decision makers (Hinkel et al., 2014). Ultimately, an uncertainty framework for coastal hazard assessment, as developed by Stephens et al. (2017), can use used to overcome these and other sources of uncertainty; however, this sort of framework is designed to guide local assessments and has not yet been expanded to the regional and global scales."

**Comment 15: I sincerely appreciate the intention to provide a Global quantification of effectiveness of DRR measures. However, I recommend that the authors analyze the effectiveness of DRRs (especially building- and community-level measures) considering local and regional dynamics with region-specific datasets and knowledge and then integrate them in such a global study.**

Response 15: We agree that it is prudent to include as much regional and local information as possible when conducting an analysis such as ours. Ideally, we would include as detailed information as possible, gathered through local physical and social surveys on various scales. Unfortunately, for a majority of the world, much of this information is not documented, if it even exists at all. Of the limited information that does exist, the issue of merging regional data in to a single dataset is not straightforward and could lead to biases depending on the data availability per region. In this sense, global datasets has the advantage of being consistent across the globe.

Still, we have attempted to capture some regionality using national construction factor corrections, different levels of relative costs between the high and low income countries, and so on. As this is a first-cut analysis, we have used these proxies and simplifications to make the analysis feasible. We foresee the potential of incorporating additional information through various means in future analyses, including the development of regional-scale agent-based models to reflect the realistic and dynamic actions of people and communities in the face of current and future flood risk. These models could then be upscaled to a larger scale for global modelling purposes to create a more realistic portrait of the effectiveness of building- and community-level measures. Certain agent-based models have already been developed on country- and continental-scales, include that of Haer et al. (2020), which considers the European context of flood risk management. To reflect this potential for future studies we will include the following text in section 3.4 of our manuscript:

> "In general, we acknowledge that the assumptions used in our global analysis do not capture a fully representative picture of what the modelled DRR measures would be in reality, especially in terms of their effectiveness, variations around the world, and potentially dynamic nature. An avenue for future research could include developing numerous regional agent-based models based on locally surveyed information to represent these dynamics and variation. A more detailed and accurate depiction of global DRR measure implementation could potentially be achieved as a result."

We do not intend local implementation for these globally modelled DRR measures based solely on this analysis; rather, our analysis serves as a starting point for a local process. In this sense, our analysis points global players and decision makers in the direction for where to act first and what options might be considered. We will add further text than what is already mentioned here throughout the manuscript highlighting this specific aim, including this passage in section 4 of the manuscript:

> "The on-the-ground design of adaptation measures requires site-specific and detailed local information, but using a globally applicable model in data-scarce regions allows end-users such as UN-affiliated organizations, the World Bank, and (inter)national adaptation strategists to prioritize actions."

We again thank the referee for their time in reviewing our manuscript and for providing useful comments that have improved the study.

References added to the revised manuscript

Haasnoot, M., Winter, G., Brown, S., Dawson, R. J., Ward, P. J., & Eilander, D. (2021). Long-term sea-level rise necessitates a commitment to adaptation: A first order assessment. *Climate Risk Management*, *34*, 100355.

Hecht, R., Meinel, G., & Buchroithner, M. (2015) Automatic identification of building types based on topographic databases – a comparison of different data sources, *International Journal Cartography*, *1*, 18–31.

Hinkel, J., Feyen, L., Hemer, M., Le Cozannet, G., Lincke, D., Marcos, M., Mentaschi, L., Merkens, J. L., de Moel, H., Muis, S., Nicholls, R. J., Vafeidis, A. T., van de Wal, R. S. W., Vousdoukas, M. I., Wahl, T., Ward, P. J., & Wolff, C. (2021). Uncertainty and bias in global to regional scale assessments of current and future coastal flood risk. *Earth's Future*, *9(7),* e2020EF001882.

Hinkel, J., Lincke, D., Vafeidis, A. T., Perrette, M., Nicholls, R. J., Tol, R. S., Marzeion, B., Fettweis, X., Ionescu, C., & Levermann, A. (2014). Coastal flood damage and adaptation costs under 21st century sea-level rise. *Proceedings of the National Academy of Sciences, 111(9)*, 3292-3297.

Huang, B., Zhao, B., & Song, Y. (2018) Urban land-use mapping using a deep convolutional neural network with high spatial resolution multispectral remote sensing imagery, *Remote Sensing for the Environment, 214*, 73–86.

Le Bars, D. (2018). Uncertainty in sea level rise projections due to the dependence between contributors. *Earth's Future, 6,* 1275–1291.

Rohmer, J., Lincke, D., Hinkel, J., Le Cozannet, G., Lambert, E., & Vafeidis, A. T. (2021). Unravelling the importance of uncertainties in global-scale coastal flood risk assessments under sea level rise. *Water, 13*(6), 774.

van Rooijen, A. A., McCall, R. T., van Thiel de Vries, J. S. M., van Dongeren, A. R., Reniers, A. J. H. M., & Roelvink, J. A. (2016). Modeling the effect of wave-vegetation interaction on wave setup. *Journal of Geophysical Research: Oceans, 121(6),* 4341–4359.

Schlumberger, J., Haasnoot, M., Aerts, J., & De Ruiter, M. (2022). Proposing DAPP-MR as a disaster risk management pathways framework for complex, dynamic multi-risk. *Iscience*, *25(10),* 105219.

Slangen, A. B., Palmer, M. D., Camargo, C. M., Church, J. A., Edwards, T. L., Hermans, T. H., Hewitt, H. T., Garner, G. G., Gregory, J. M., Kopp, R. E., Malagon Santos, V., & van de Wal, R. S. (2023). The evolution of 21st century sea-level projections from IPCC AR5 to AR6 and beyond. *Cambridge Prisms: Coastal Futures*, *1*, e7.

Stephens, S. A., Bell, R. G., & Lawrence, J. (2017). Applying principles of uncertainty within coastal hazard assessments to better support coastal adaptation. *Journal of Marine Science and Engineering*, *5(3),* 40.

Ward, P. J., Jongman, B., Salamon, P., Simpson, A., Bates, P., De Groeve, T., Muis, S., Coughlan de Perez, E., Rudari, R., Trigg, M. A., & Winsemius, H. C. (2015). Usefulness and limitations of global flood risk models. *Nature Climate Change*, *5(8),* 712-715.

---

## Author Response (AR2)

**REFEREE 1**

**Comment 1: The manuscript by Mortensen et al. employs a global flood model to estimate the effectiveness of a range of disaster risk reduction (DRR) measures in limiting future flood risk, focusing on potential direct impacts. In their assessment, the authors use the constant relative-risk objective, with respect to the regional Gross Domestic Product (GDP). The paper is well structured, clearly written, the results are presented in detail and the limitations are acknowledged and outlined in a clear manner. The study produces interesting results regarding the effectiveness of the different measures in reducing risk and regarding the cost-benefit ration of these measures and are therefore interesting for developing regional strategies to manage coastal flood risk.**

**I would recommend the study for publication – nevertheless, I would like to post some comments that I believe need to be addressed or may be useful for the authors to improve the paper:**

Response 1: We thank the referee for their assessment of our manuscript, and appreciate the general feedback given for improvement. We specifically reply to the points raised below.

**Comment 2: I am unsure why the authors refer to the measures that they are exploring as DRR measures. I do understand that floods can be disastrous but, if I am not mistaken, the authors are not assessing risk based only on high-impact low-probability events that could lead to disasters; rather, they estimate flood probabilities integrating over a range of return periods (ranging from events with 2-year return period, which hardly constitute disasters, to events with 1000-year return period). To my knowledge, these measures are usually referred to in the literature as coastal adaptation measures or grouped under the IPCC coastal adaptation typology categories. I find this potentially confusing and would suggest the authors to either explain clearly why the term DRR is used or refer to the IPCC terminology for coastal adaptation.**

Response 2: The authors believe that coastal adaptation, disaster risk reduction, and climate change adaptation are all interrelated terms. We will include the following text in the second paragraph of section 1 of the manuscript to clarify our decision to use disaster risk reduction:

> "Forward looking disaster risk reduction (i.e. prospective disaster risk management) that examines potential future risks under climate change scenarios specifically also examines future risk, as does climate adaptation. Indeed, there is a call in the policy and science literature to bridge the silos between these domains (UNDRR, 2020). In our analysis, because the measures as implemented to protect against a quantifiable return period of inundation

in the future, we use the term DRR to refer to any actions taken to address changes in coastal flood risk."

An extra point of point of clarification in response to the referee – our framework does not only assess impacts from large disasters. We calculate impacts for return periods between 2 years (high frequency) and 1000 years (low frequency) and calculate risk by integrating across impacts for all return periods.

**Comment 3: The authors use a GIS-based inundation model, considering attenuation of water levels and, if I am not mistaken, waves. It is however not clear to me how waves have been accounted for in the total water level. Could the authors please clarify? Also, I would assume that the wave values that the authors are using refer to offshore waves; or does the model propagate waves to the near shore? (and how near is the "near shore", since wave height will change considerably as waves approach the coast)**

Response 3: The hazard dataset that we have used in our analysis does not include waves or wind-related surface interactions; a simple bathtub model with dynamic resistance is used in our modelling scheme. Instead, we account for wave height in the implementation and performance of DRR measures that are hazard specific, namely dykes and levees and foreshore vegetation. For dyke height increases, current dyke heights are derived for coastal segments and perpendicular coast-normal transects (766,034 transects in total). For each transect, bed levels are constructed and, subsequently, hydrodynamic conditions and near-shore wave attenuation are derived. Lastly, the resulting sea water levels are translated into dike heights. For further details of this methodology, refer to Tiggeloven et al. (2020) and van Zelst et al. (2020). A similar methodology is used to account for wave height reduction resulting from foreshore vegetation, with further details found in Tiggeloven et al. (2022). We will revise the final paragraph of section 2.1.1 of the manuscript as follows:

> "We follow the Peak Over Threshold (POT) method and we fit the Generalized Pareto Distribution (GPD) on the peaks that exceed the 99th percentile surge level. From there we derive estimated sea levels for various return periods. These computed sea levels are then used as input for a GIS-based inundation model using the MERIT DEM (Yamazaki et al., 2018), as described in Tiggeloven et al. (2020), to simulate the inundation. This is a static flood model that simplifies all dynamic processes into a single attenuation factor of the water levels over land (Vafeidis et al., 2019), resulting in a simple bathtub model with static forcings instead of a more complex dynamic inundation model framework. The flood maps do not include wind-waves or future changes in waves and storminess. Rather, (nearshore) waves are accounted for in calculation of the hazard-specific DRR measures effect on inundation levels, discussed further in sections 2.2.1 and 2.2.2 below.

Results of RCP6.0, an intermediate climate change scenario (O'Neill et al., 2017), are explored here in the main text. Additional RCPs are available in the supplement."

With this revision to section 2.1.1, the text describing the method of accounting for waves in section 2.2.1 is now more clear. In section 2.2.2 of the manuscript, the following sentences will be added to clarify how waves are accounted for in modelling foreshore vegetation effects on coastal flood hazard reduction:

"Similar to Tiggeloven et al., (2022), here wave conditions are derived from the ERA-Interim (Dee et al., 2011) reanalysis using a peak-over-threshold approach. To determine the wave attenuation over a foreshore and the resulting significant wave height relevant for the flood defense on a transect, we search an existing lookup-table (van Rooijen et al., 2016) of hydrodynamic numerical modelling results for combinations of foreshore slopes, vegetation covers and hydrodynamic conditions (van Zelst et al., 2021). These searched wave heights are modelled at regular intervals along a steady slope, both with and without salt marsh or mangrove vegetation. Wave angle of incidence is assumed coast normal. Wave attenuation along the vegetated coastlines is determined based on the closest match between the derived transects characteristics and look-up table results."

**Comment 4: Although the authors outline very clearly the limitations of the study, there is hardly any discussion on uncertainty and how this is addressed – where do the main uncertainties in the results stem from? I guess it would be too much to ask the authors to conduct an uncertainty analysis but there is substantial literature regarding flood risk assessments and the authors should at least discuss this issue.**

Response 4: We thank the referee for this valid concern. We will incorporate several lines of text describing the uncertainties that might arise from our analysis. The lines will be incorporated into section 3.4, with the title of this section being revised to reflect our addition:

"Uncertainty in our analysis originates from several sources, including data inputs and modelling assumptions. This is also discussed in other global-scale coastal flood risk assessment literature. With regard to scenario uncertainty, Rohmer et al. (2021) state that adaptation costs are most sensitive to RCP used, while EAD is more sensitive to SSP. RCPs ultimately drive sea-level rise projections, which are also based on thermal expansion, global surface air temperature and ocean dynamic sea level from the Coupled Model Intercomparison Project 5 (CMIP5) with IPCC AR5 estimations of ice and land water contributions complemented with the reevaluation of Antarctic contribution from SROCC. The uncertainties are combined based on the probabilistic model described in Le Bars (2018).

Tiggeloven et al. (2020) sees the largest sensitivity for global adaptation costs stemming from sea-level rise. Indeed, the largest source of scenario uncertainty, according to Hinkel et al. (2021), relates to future coastal adaptation scenarios, which can influence future coastal flood risk by factors of 20.0–26.7. It is this exact source uncertainty that we explore with our analysis by employing several DRR measures, reaffirming that future coastal flood risk depends greatly on which action is taken by decision makers (Hinkel et al., 2014). Ultimately, an uncertainty framework for coastal hazard assessment, as developed by Stephens et al. (2017), could be used to overcome these and other sources of uncertainty such as data input uncertainty (e.g., DEM and exposure maps); however, this sort of framework is designed to guide local assessments and has not yet been expanded to the regional and global scales."

**Comment 5: Following my previous point, uncertainty (in e.g. socio-economic development) is often addressed with the use of scenarios. The authors use only one scenario combination (SSP2-RCP6.0), which is a middle of the road scenario; I am unsure what the value of this is since it gives us practically no information about the potential range of uncertainty. In this case, either a second scenario should be used or the authors should rather opt for a high- or low-end scenario which would indicate the upper or lower boundaries. Of course, there is value in comparing the different measures, however, in a different scenario combination results could look very different.**

Response 5: Indeed, as the referee mentions, our primary focus for this analysis was the effect of the DRR measures on future coastal flood risk impacts. In addition to the aforementioned text detailing potential sources of uncertainty, we intend to include certain scenario combinations and subsequent results in the supplement of the manuscript. The combinations SSP2-RCP2.6, SSP2-RCP4.5, and SSP2-RCP8.5 are therefore included in the supplement. The following text will be included in section 3.3 of the manuscript:

"While here we only present the results of SSP2-RCP6.0, additional results are available for other SSP-RCP combinations in the supplement, namely combinations with varying RCPs. These additional results show that while the overall magnitude of increases to future risk remain substantial regardless of combination, larger EAD values can be expected with higher-end RCPs. Additionally, we see the effectiveness of certain DRR measures, specifically foreshore vegetation and zoning restrictions, decreasing with higher-end RCPs."

Specific value ranges will be added to Table 2 of the manuscript to reflect this textual addition, thus representing the range of potential outcomes under various climate scenarios. In our analysis, we specifically wanted to focus on the impact of climate change on the future of DRR, and not necessarily socioeconomic development or other sources of uncertainty. We chose this route, among other reasons, because of

the large amount of attention currently given on the global stage to concerted efforts in reducing the amount of warming that occurs on Earth due to the impact that this warming has on sea-level rise (Slangen et al., 2022). This notion is seen in IPCC reports and annual COP meetings. Nevertheless, we will include the following sentence in section 3.4 of the manuscript:

> "While here we chose to focus only on the impact of climate change (RCPs) on the future of DRR, further research could integrate different SSP storylines to investigate socio-economic change rather than the effects of climate change as we do in the paper."

**Comment 6: I understand the need for a no-measures assessment. However, I believe that it should be clearly pointed out that this is just a theoretical exercise since, in reality, there will be a response to flooding and adaptation will take place in one form or another, at some point in the century.**

Response 6: We agree with the referee and will add the following sentences in the opening of section 3.1 of the manuscript to clarify this point:

> "The need for a no DRR action assessment stems from the theoretical exercise of determining benefits achieved by implementing any given DRR measure. In reality, a future with no DRR action whatsoever is highly improbable. Communities increasingly at risk to coastal flooding will react to the changing conditions. Still, here we quantify this no DRR action scenario as the basis of how much reduction to coastal flood risk is required and is possible."

**Comment 7: Foreshore vegetation can be partly effective in reducing flood risk – however, a high-end event would destroy foreshore vegetation, thus limiting its protective effects for the years to come. I assume that this has not been considered, I however believe that it would be useful to discuss.**

Response 7: We assume that all measures – foreshore vegetation and otherwise – are implemented and do not experience any failure below the threshold of the protection standard provided. In reality, high-end events could reduce the effectiveness of any risk reduction measure, thus necessitating their repair or replacement.

If we were to include actual benefit losses and subsequent replacement cost calculations within our analysis due to destruction of foreshore vegetation, we would expect overall risk levels would slightly increase during the few years needed for the foreshore vegetation to rejuvenate or be replaced. This could be very interesting to investigate in future work, but here it serves as a limitation as we do not focus on pathway evolution through time, but rather on one point in the future (in our case 2080). Certain studies, such as Haer et al. (2020) and Schlumberger et al. (2022) look at time dynamics of adaptive action, albeit at much smaller scales. This timesensitive component of adaptation was not within the scope of this manuscript, but is interesting to explore further.

We will include this important point raised by the referee as a topic of discussion in section 3.4 of the manuscript with the following text:

> "While here we have assumed our DRR measures do not experience any failure below the threshold of provided protection standards, violent storm events could in reality partially damage or destroy the DRR measures, in particular sensitive ones such as foreshore vegetation. This limitation results in the potential underestimation of costs of (re)implementation and overestimation of benefits provided by measures if they were to experience such failures."

**Comment 8: My last point is a suggestion: based on my experience, many of the differences in global flood impact assessments stem from the calculation of the floodplains. I would personally find it useful if the authors would make their floodplains freely available (not only upon request as this usually does not work) so that others can use them to produce estimates that are comparable. I believe there could be a lot of added value for the research community if everyone conducting global or continental impact assessments made their floodplains openly available.**

Response 8: We agree with the referee's comment on the importance of making data publicly available. In fact, the hazard data used in the most prior application of GLOFRIS is already publicly available via the World Resource Institutes' Aqueduct webtool (www.wri.org/publication/aqueduct-floods-methodology). We intend to make the updated hazard extent maps created explicitly for our analysis available for public use under creative commons licensing.

**Comment 9: I hope my comments help the authors to strengthen this very good manuscript.**

Response 9: We again thank the referee for their time in reviewing our manuscript and for providing useful comments that have improved the study.

**REFEREE 2**

**Comment 10: In this study, the authors estimate the effectiveness of DRR measures for coastal flooding and provide sub-national risk estimates. This is a complex topic given the dynamics in hazard, exposure and vulnerability components. DRR measures are very important for reducing flood risk. Firstly, thank you for addressing this important aspect in flood risk management.**

Response 10: We thank the referee for their assessment of our manuscript, and appreciate the general feedback given for improvement. We specifically reply to the points raised below.

**Comment 11: The authors mention that one of the novel aspects of the study is the global scale of analysis. Unfortunately, I have major concerns regarding the assumptions behind the risk computation and hence, the overall take away from this study.**

Response 11: We indeed assert that our global-scale analysis is novel in that it examines a previously unexplored set of DRR measures all within the same global flood risk model (GFRM), one of several research gaps that was identified by Ward et al. (2015). This article, which discusses the usefulness and limitations of GFRMs, was erroneously excluded from our list of references and has been textually inserted where appropriate, specifically in discussions of uncertainty and limitations of our analysis (see below).

**Comment 12: I like the concept of risk constant. However, many other assumptions are quite vague to generalize. The possibility to implement DRR measures and their effectiveness to reduce risk are very diverse across regions and countries. For example, the assumption such as a constant % of dry-proofed area and urban cell composition are too simplified for a cost analysis and could be wrong for many regions. The same with the generalized costs of zoning. The authors do mention that as a limitation, however, it is a significant limitation that questions the credibility and usability of the results presented.**

Response 12: The referee brings up a fair critique of some of the assumptions used in our global modeling framework. We will add the following text to clearly state these limitations in section 3.4 of the manuscript:

> "Several assumptions are made in the implementation of our DRR measures. For example, we assume the percentage of occupancy type per grid cell to be the same for all locations, whilst in reality it is spatially heterogeneous. We also assume building density per occupancy type. An improvement to our analysis could be made by using machine learning to improve accuracy of urban land cover and building types (Hecht et al., 2015; Huang et al., 2018). Furthermore, while we have assumed a rapid adoption of DRR measures and full

effectiveness/uptake, timing and rate of a commitment to adaptation varies per country (Haasnoot et al., 2021), which we do not consider here."

On the credibility and usability we want to stress the purpose of global analysis versus local analysis, as also outlined in Ward et al. (2015) on the advantages and disadvantages of global scale analysis. We argue that global analyses such as ours support dialogue with stakeholders, including policy and decision makers, and identify priority regions for action. Conducting global-scale risk analysis for disaster risk reduction is essential for gaining a comprehensive understanding of interconnected risks, addressing transboundary challenges, facilitating comparative analysis, promoting collaboration, and developing effective strategies to prevent and mitigate global risks. We intend our framework to be used to highlight potential savings (in the form of expected damage reductions) through strategies which increase DRR at the sub-national scale.

However, as the referee also highlights, when moving towards implementation of individual DRR measures identified by this and other global studies, detailed studies should be performed using local models and data. This sentiment is captured by several other proposed textual additions to the manuscript that are explicitly spelled out in this response (see responses 4, 14, and 15).

**Comment 13: I see that two out of the four DRR measures – dikes and foreshore vegetation are part of the previous work done by the co-authors (as cited in this manuscript). The new findings are the effects of dry-proofing and zoning (please clarify if I am missing something here).**

Response 13: We will add the following text to section 1 of the manuscript to clarify the novelty of our analysis:

> "In our analysis, we have developed and modelled dry-proofing and zoning restrictions as DRR measures, which has never been done before on the global scale. We have also incorporated previously unconsidered costs for foreshore vegetation, namely mangrove restoration costs. To fully compare these new findings, the flood risk impact reduction potential of dykes and levees as well as foreshore vegetation are also recalculated using new hazard and exposure data that were developed explicitly for this analysis."

**Comment 14: I strongly believe that there is a definite need to motivate the implementation of DRR measures. However, the generalized assumptions made in the study without considering local processes make the risk numbers at the Global level questionable. Also, the authors have not provided uncertainty ranges or any sort of validation for any of the reported values (e.g. EAD and EAAP; risk-reduction due to measures).**

Response 14: Thank you for this excellent suggestion to include uncertainty ranges. We will include additional scenarios combinations in the supplement and text within the main body of the manuscript to address scenario uncertainty. In our analysis, we specifically wanted to focus on the impact of climate change on the future of DRR, and not necessarily socioeconomic development or other sources of uncertainty. The combinations SSP2-RCP2.6, SSP2-RCP4.5, and SSP2-RCP8.5 have therefore been included in the supplement. The following text will be inserted in section 3.3 of the manuscript:

> "While here we only present the results of SSP2-RCP6.0, additional results are available for other SSP-RCP combinations in the supplement, namely combinations with varying RCPs. These supplementary results show that while the overall magnitude of increases to future risk remain substantial regardless of the SSP-RCP combination, larger EAD values can be expected with higher-end RCPs. Additionally, we see the effectiveness of certain DRR measures, specifically foreshore vegetation and zoning restrictions, decreasing with higher-end RCPs."

Specific value ranges will be added to Table 2 of the manuscript to reflect this textual addition. To discuss further potential sources of uncertainty, we will add text to section 3.4 of the manuscript:

> "Uncertainty in our analysis originates from several sources, including data inputs and modelling assumptions. This is reflected in other global-scale coastal flood risk assessment literature. With regard to data inputs, Rohmer et al. (2021) state that adaptation costs are most sensitive to RCP used, while EAD is more sensitive to SSP. This notion is supported by Tiggeloven et al. (2020), which sees a majority of the sensitivity for global adaptation costs stemming from sea-level rise. Indeed, the largest source of uncertainty, according to Hinkel et al. (2021), relates to future coastal adaptation scenarios, which can influence future coastal flood risk by factors of 20.0–26.7. It is this exact source uncertainty that we explore with our analysis by employing several DRR measures, reaffirming that future coastal flood risk depends greatly on which action is taken by decision makers (Hinkel et al., 2014). Ultimately, an uncertainty framework for coastal hazard assessment, as developed by Stephens et al. (2017), can use used to overcome these and other sources of uncertainty; however, this sort of framework is designed to guide local assessments and has not yet been expanded to the regional and global scales."

**Comment 15: I sincerely appreciate the intention to provide a Global quantification of effectiveness of DRR measures. However, I recommend that the authors analyze the effectiveness of DRRs (especially building- and community-level measures) considering local and regional dynamics with region-specific datasets and knowledge and then integrate them in such a global study.**

Response 15: We agree that it is prudent to include as much regional and local information as possible when conducting an analysis such as ours. Ideally, we would include as detailed information as possible, gathered through local physical and social surveys on various scales. Unfortunately, for a majority of the world, much of this information is not documented, if it even exists at all. Of the limited information that does exist, the issue of merging regional data in to a single dataset is not straightforward and could lead to biases depending on the data availability per region. In this sense, global datasets has the advantage of being consistent across the globe.

Still, we have attempted to capture some regionality using national construction factor corrections, different levels of relative costs between the high and low income countries, and so on. As this is a first-cut analysis, we have used these proxies and simplifications to make the analysis feasible. We foresee the potential of incorporating additional information through various means in future analyses, including the development of regional-scale agent-based models to reflect the realistic and dynamic actions of people and communities in the face of current and future flood risk. These models could then be upscaled to a larger scale for global modelling purposes to create a more realistic portrait of the effectiveness of building- and community-level measures. Certain agent-based models have already been developed on country- and continental-scales, include that of Haer et al. (2020), which considers the European context of flood risk management. To reflect this potential for future studies we will include the following text in section 3.4 of our manuscript:

> "In general, we acknowledge that the assumptions used in our global analysis do not capture a fully representative picture of what the modelled DRR measures would be in reality, especially in terms of their effectiveness, variations around the world, and potentially dynamic nature. An avenue for future research could include developing numerous regional agent-based models based on locally surveyed information to represent these dynamics and variation. A more detailed and accurate depiction of global DRR measure implementation could potentially be achieved as a result."

We do not intend local implementation for these globally modelled DRR measures based solely on this analysis; rather, our analysis serves as a starting point for a local process. In this sense, our analysis points global players and decision makers in the direction for where to act first and what options might be considered. We will add further text than what is already mentioned here throughout the manuscript highlighting this specific aim, including this passage in section 4 of the manuscript:

> "The on-the-ground design of adaptation measures requires site-specific and detailed local information, but using a globally applicable model in data-scarce regions allows end-users such as UN-affiliated organizations, the World Bank, and (inter)national adaptation strategists to prioritize actions."

To further clarify that this is a theoretical exercise and not one based on observed values, the concept of the effectiveness metric has been renamed as efficacy metric, defined as the performance of any given DRR measure under ideal and controlled circumstances.

We again thank the referee for their time in reviewing our manuscript and for providing useful comments that have improved the study.

References added to the revised manuscript

Haasnoot, M., Winter, G., Brown, S., Dawson, R. J., Ward, P. J., & Eilander, D. (2021). Long-term sea-level rise necessitates a commitment to adaptation: A first order assessment. *Climate Risk Management*, *34*, 100355.

Hecht, R., Meinel, G., & Buchroithner, M. (2015) Automatic identification of building types based on topographic databases – a comparison of different data sources, *International Journal Cartography*, *1*, 18–31.

Hinkel, J., Feyen, L., Hemer, M., Le Cozannet, G., Lincke, D., Marcos, M., Mentaschi, L., Merkens, J. L., de Moel, H., Muis, S., Nicholls, R. J., Vafeidis, A. T., van de Wal, R. S. W., Vousdoukas, M. I., Wahl, T., Ward, P. J., & Wolff, C. (2021). Uncertainty and bias in global to regional scale assessments of current and future coastal flood risk. *Earth's Future*, *9(7),* e2020EF001882.

Huang, B., Zhao, B., & Song, Y. (2018) Urban land-use mapping using a deep convolutional neural network with high spatial resolution multispectral remote sensing imagery, *Remote Sensing for the Environment, 214*, 73–86.

Le Bars, D. (2018). Uncertainty in sea level rise projections due to the dependence between contributors. *Earth's Future, 6,* 1275–1291.

van Rooijen, A. A., McCall, R. T., van Thiel de Vries, J. S. M., van Dongeren, A. R., Reniers, A. J. H. M., & Roelvink, J. A. (2016). Modeling the effect of wave-vegetation interaction on wave setup. *Journal of Geophysical Research: Oceans, 121(6),* 4341–4359.

Schlumberger, J., Haasnoot, M., Aerts, J., & De Ruiter, M. (2022). Proposing DAPP-MR as a disaster risk management pathways framework for complex, dynamic multi-risk. *Iscience*, *25(10),* 105219.

Slangen, A. B., Palmer, M. D., Camargo, C. M., Church, J. A., Edwards, T. L., Hermans, T. H., Hewitt, H. T., Garner, G. G., Gregory, J. M., Kopp, R. E., Malagon Santos, V., & van de Wal, R. S. (2023). The evolution of 21st century sea-level projections from IPCC AR5 to AR6 and beyond. *Cambridge Prisms: Coastal Futures*, *1*, e7.

Stephens, S. A., Bell, R. G., & Lawrence, J. (2017). Applying principles of uncertainty within coastal hazard assessments to better support coastal adaptation. *Journal of Marine Science and Engineering*, *5(3),* 40.

UNDRR (2020). Integrating Disaster Risk Reduction and Climate Change Adaptation in the UN Sustainable Development Cooperation Framework, UN Office for Disaster Risk Reduction.

UNFCCC (2014). Report of the Conference of Parties on its nineteenth session, held in Warsaw from 11 to 23 November 2013, UN Framework Convention on Climate Change.

Ward, P. J., Jongman, B., Salamon, P., Simpson, A., Bates, P., De Groeve, T., Muis, S., Coughlan de Perez, E., Rudari, R., Trigg, M. A., & Winsemius, H. C. (2015). Usefulness and limitations of global flood risk models. *Nature Climate Change*, *5(8),* 712-715.

**REFEREE 3**

**Comment 16: The manuscript by Mortensen et al. presents a global assessment of the effectiveness and economic performance of four disaster risk reduction measures in reducing the risk of current and future coastal flooding. I find the article to be of very good quality, well written, with an extensive methodology and with good and valuable results. The findings presented here are of great interest to the coastal research community and coastal managers. In this sense, my overall impression is positive, and I recommend the publication. I nonetheless do have some comments that may help to improve the manuscript.**

Response 16: We thank the referee for their assessment of our manuscript, and appreciate the general feedback given for improvement. We specifically reply to the points raised below.

**Comment 17: Usually, when discussing disasters, we consider the worst scenarios. In this case, an analysis and comparison, at least with RCP 8.5, should be included in the study. You mentioned (Lines 130 and 380, for example) that additional results are available for other RCP/SSP combinations in the supplement. However, there is no other supplementary material but a spreadsheet. One of the previous reviewers commented about the importance of this analysis, and I agree with them.**

Response 17: The sentiment that the referee touches upon here is one that has been considered at length by the co-authors throughout the entire analysis and writing process. Originally, we chose to focus the main text of the manuscript solely on one RCP/SSP combination (in our case, RCP6.0 / SSP2) because we wanted the manuscript to highlight the novelty of incorporating several modelled DRR measures into one flood risk assessment framework, and the subsequent comparisons of various performance-based indicators (e.g., BCR and efficacy). We did not want the focus of our paper to be on the uncertainty of future climate and population projections, which indeed can serve as a rich component for assessments of this variety, such as that of Tiggeloven et al. (2020) and Hinkel et al. (2021).

We now recognize, though, as the referee suggests here (as well as the first referee in comment 5), that our study as originally presented perhaps missed an important analytical facet by focusing our main text on solely one RCP/SSP. By not incorporating ranges of results and focusing only on a "Middle of the Road" scenario, our manuscript neglects to acknowledge the increasing likelihood of a worst case scenario, which may give decision makers reading our manuscript the wrong message. In the interest of incorporating this information as succinctly as possible into the manuscript, the authors have decided to include additional text about two RCP/SSP combinations. We have selected RCP 2.6 / SSP1 and RCP 8.5 / SSP3, which represent low-end and high-end scenarios, respectively, in comparison to RCP 6.0 / SSP2.

Examples of text added to the manuscript include in section 3.1:

> "...Without any DRR action (i.e., only maintaining the height of current protection structures), EAD is projected to increase by over a factor of 300 and global EAAP is projected to roughly triple by 2080 (Table 1). By the year 2080, we estimate global EAD will be over $1.3 trillion, while global EAAP will exceed 11.5 million people. When considering low- and high-end scenarios, EAD ranges between $650 million and nearly $2 trillion, while EAAP ranges between 9 million and 18 million... Of particular concern in these lower-income regions is the magnitude of range, and therefore uncertainty, of potential impacts depending on the RCP / SSP combination considered. For example, we estimate EAD in Sub-Saharan Africa could range between $196 million and $834 million. If one compares a range of this magnitude to that of a higher-income region, such as Europe & Central Asia (with an estimated EAD range of $15 million and $25 million), the economic imperative of comprehensive DRR planning on the global scale is underscored.."

and in section 3.3:

> "... All globally aggregated NPVs are also positive, with the highest exhibited by dykes and coastal levees, followed by dry-proofing; however, NPV varies greatly for dykes and coastal levees as costs of heightening these structural features outpaces the benefits of doing so under high-end scenario. In general, BCRs, NPVs, and efficacy scores for the DRR measures increase under the low-end scenario (RCP 2.6 / SSP1), and decrease under the high-end scenario (RCP 8.5 / SSP3)."

Although we still intend to focus the majority of our manuscript's narrative on the results RCP6.0/SSP2, the inclusion these two additional high- and low-end RCP / SSP combinations vacates much of our original response to comment 5; the proposed passages of text referenced in that response have been removed from our manuscript to reflect the response to this comment.

Finally, we apologize for misinterpreting the term "supplementary material" as defined by NHESS (i.e., "additional figures and tables, highly detailed and specific technical information, user manuals, maps, or very large images, etc."). In reality, we have always intended to deposit the results which are not explored in the main text as a dataset (along with the datasets used as input) in a FAIR-aligned data repository. We regret any confusion caused by our improper use of terminology. We mention this fact several times throughout the manuscript. We hope those reading the manuscript who wish to further investigate the implications of different RCP/SSP scenarios.

We are grateful to the referee for pushing to include this important component of the analysis, which in turn has made our study even more relevant and stronger than before.

**Comment 18: Also, you mentioned adding at least ranges to Table 2, but again, I haven't seen it. Since you are dealing with a SLR scenario, you need to address uncertainty ranges. This is a major review that you must address.**

Response 18: This was an unintended omission on our part in the finalization of the first round of the review process. We have included the new Table 2 (including high- and low-end scenario ranges) in the revised manuscript.

**Comment 19: Still concerning, looking at the results of your work, it appears that building dykes is the best solution for the entire world. And we know that hard engineering solutions can cause more problems than solutions. I know it is not the aim of your work, but you should address this discussion. Otherwise, your global assessment, in which you argue it "supports dialogue with stakeholders, including policy and decision-makers, and identifies priority regions for action", can be misinterpreted.**

Response 19: These results initially puzzled the co-authors, but when considering how the DRR measures have been implemented in the modelling framework they make sense. In the cases of dry-proofing, zoning restrictions, and foreshore vegetation, there are spatial limitations – there is only so much area that can be floodproofed, restricted, or restored. Meanwhile, dykes and levees have been modelled without a physical "upper-limit", meaning that any height increase necessary to achieve the risk target (as described in 2.1.4) is possible up to, but not including, the current 1000-year inundation level. A restriction could have been incorporated into our analysis by setting an arbitrary limit of dyke and levee improvements (e.g., no more than X meters of dyke raising allowed or up to XX-year protection standard) to address this mismatch of DRR measure implementation, however this could have resulted in unfair constraints placed on areas with little to no estimated existing protection infrastructure in place.

We have added text to section 2.2 to address this nuanced comment:

> "In the case of foreshore vegetation, zoning restrictions, and dry-proofing, spatial limits exist for the amount of action that can be taken – only so much area can be restored to mangroves, restricted from urbanisation, or dry-proofed, depending on the context. Meanwhile, we have modelled dykes and coastal levees with no physical limitation, such as a height limitation or a maximum protection standard provided (aside from the theoretical limit of the 1000-year flood event)."

**Comment 20: I also noticed that, although extensive, your methodology lacks details, and the authors always refer to other authors' works. You made several assumptions, but you don't describe all of them (which I agree is almost impossible). In this sense, the methodology must be very clear for the reader understand exactly what was used in this work. You could still detail some**

**parts, such as how EAD and EAAP are calculated. I had to read Tiggeloven et al. (2020) to understand your methodology.**

Response 20: While it is difficult to succinctly describe all of the assumptions of our study, we have feel we have, by and large, captured the essence of our work in the text while referring readers to relevant sources for additional information. We see that we neglected to include a description of EAD and EAAP integral calculations, and agree with the referee that the source for this information should be included explicitly as opposed to pointed to via a different study using a similar methodology, in this case Tiggeloven et al. (2020) which uses Meyer et al. (2009). We have added the following text to section 2.1 of our manuscript:

> "After calculating the impacts for the different return periods, both EAD and EAAP is calculated by taking the integral of the exceedance probability–impact curve (Meyer et al., 2009)."

**Comment 21: Also, if feasible, include a figure with the methodological framework, including the data you used for each estimation model. Figure 1 is good but is very general. There is a lot of information in your extensive methodology that could be more easily visualized in a figure.**

Response 21: We have attempted to summarize the fundamental components of our methodology into Figure 1 without creating a figure that was unnecessarily busy or crowded. With the clarified text from above, as well as a further expanded figure caption (see response 33), we feel the need for an additional figure in the methodology is assuaged.

**Comment 22: It is not clear to me why or how some aspects important to inundation are not included in the risk estimation but are included in the risk reduction, for example, waves and subsidence. Could you discuss the implications of it? Including waves and subsidence in some parts of the world can change drastically the risk estimation. Other important discussions to address are coastal squeeze due to sea level rise (related to foreshore vegetation) and the implications of not considering zoning restrictions in highly urbanized areas (for example, not considering retreat in the model).**

Response 22: First, for clarification, subsidence is included in our risk estimation (see response 28). Indeed, we notice large localized increases to flood hazard by accounting for this phenomenon in our analysis.

The referee raises several worthy and very interesting points of discussion (some of which are briefly touched upon already in section 3.4 of the manuscript). While subsidence is considered in our analysis, waves were not included in the hazard dataset due to technical limitations at the time of the analysis. Waves were subsequently only considered in the modelling of the dykes and levees DRR measure,

as we deemed waves relevant for dyke overtopping scenarios which impact resulting protection standards. As the referee alludes to, neglecting wave action could result in a slight underestimation of actual coastal flood risk, and perhaps an overestimation of the other DRR measures modelled in our analysis.

As for the other discussion points raised, we agree with the referee that 1) coastal squeeze due to sea level rise *and* further urbanization can hinder the usefulness of foreshore vegetation as a DRR measure and 2) managed retreat must be considered as an adaptation option by decision makers in certain contexts. While we briefly mention the latter, we have neglected to include the former in the text and have added a sentence in section 3.4 of the manuscript:

> "… in reality, violent storm events could partially damage or destroy the DRR measures, in particular sensitive ones such as foreshore vegetation. Meanwhile, coastal squeeze caused by sea-level rise and urbanisation could cause further reductions of effectiveness of certain DRR measures (Torio & Chmura, 2013). These limitations result in the potential underestimation…"

**Comment 23: Finally, I agree with reviewer 1 to make your data (and code) freely and readily available, and not only "upon request".**

Response 23: The datasets used for analysis as well as results will be publicly available. While we can appreciate the availability of open-access code for many scientific purposes, GLOFRIS will remain available upon request only for purpose of proprietary protection.

**Comment 24: Line 114 – What is the reason for choosing only RCP 6.0? Please include one sentence justifying your answer. Same for the choosing of SSP 2 in line 140.**

Response 24: We selected the combination of RCP 6.0 and SSP 2 after a long period of discussion and consultation amongst the co-authors. Ultimately, we decided to select a "Middle of the Road" scenario to create what could be considered a realistic estimate of benefits and costs to be achieved through implementation of various DRR measures. As we have geared our manuscript towards decision and policymakers, we did not want to present worst-case scenario monetary and human impacts, but rather realistic values that they would have a more likely chance of encountering in their adaptation and planning decisions. This is an approach that has been taken up by governmental agencies such as the Dutch Environmental Assessment Agency (PBL). The following sentence has been included into section 2.1 of the manuscript:

> "RCP 6.0 / SSP2 is chosen here as a climate change adaptation scenario due to its representation of an intermediate level of climate change mitigation, moderate socioeconomic challenges, and a balanced emphasis on economic

development, social equity, and environmental sustainability, thus presenting a realistic scenario for decision and policymakers to consider."

In light of the inclusion of two additional RCP/SSP scenarios in our analysis, we trust this straightforward justification will suffice.

**Comment 25: Still, about the RCP 6.0, you don't mention which percentile you used, if it was the 50th or the range 5th-95th percentile. Was only one value or a range of probabilistic outcomes? You examine the results for 2080, but equally important is to address the uncertainties related to it.**

Response 25: In section 2.1.1 we have added the following text to clarify our methodology:

"Results of RCP 6.0 (50$^{th}$ percentile), an intermediate climate change scenario, are explored here in the main text. Results using additional RCPs are available in the supplement."

**Comment 26: Still about hazards, are you using the same datum for the entire global analysis? How are you dealing with different data sources, which are not necessarily in the same datum? Or are they? Please include a sentence about it.**

Response 26: The hazard maps are globally calculated for mean sea level. We then apply a spatially varying offset between mean sea level according to the FES2012 model and the datum used by the terrain model MERIT (EGM96) to ensure that the zero datum of our terrain and our extreme sea levels from GTSR are the same. This clarification has been concisely added to the text in Section 2.1:

"… using the MERIT (EGM 96) DEM (Yamazaki et al., 2018) with spatially varying offset between mean sea level according to the FES2012 model, as described in Tiggeloven et al. (2020), to simulate global inundation using the same datum."

**Comment 27: Line 144 – The built-up area used refers to what exactly? Does it include roads and parks? Please include the meaning of built-up areas.**

Response 27: We have included the following text to further clarify our definition of built-up areas:

"Here, built-up area refers to all kinds of built environs (e.g., buildings) and artificial surfaces (e.g., paved surfaces)."

**Comment 28: Line 170 and 190 – Do you also include subsidence when calculating the inundation maps? I can see you do when calculating dyke height. But subsidence is essential in general for coastal flooding, after all, it can**

**accelerate the process. It's not clear in the text if you include it or not. If not, please explain why you only use it for dyke heights. Or why you haven't used it for the maps.**

Response 28: We agree with the referee in their point raised: subsidence is an essential process to include when considering future coastal flood risk. Indeed, we include subsidence when calculating the inundation maps. We have clarified this in the text with the following lines in section 2.1.1:

> "Similarly to Tiggeloven et al. (2020), we include subsidence in the estimation of future coastal flood risk. These subsidence rates are determined by the SUB-CR model (Kooi et al., 2018). Because subsidence is a highly regional phenomenon, rates of subsidence are applied to specified locations of Kooi et al. (2018) at a spatial resolution of 30″ x 30″ (which is a spatial interpolation based on the original spatial scale of 5′ x 5′)."

**Comment 29: Line 91 – How were EAD and EAAP calculated? I've noticed there is no information about this in the methodology. Could you at least provide the reference for this method?**

Response 29: With our response to comment 20, this issue is resolved. Specifically, we have included Meyer et al. (2009) as a new reference.

**Comment 30: Line 300 – What is causing the difference between your and Tiggeloven's EAD results for no DRR action? Your result is twice what they had. You have different scenarios, but both are "middle of the road". Is it the only explanation for duplicating the results? What else is influencing?**

Response 30: We are extremely grateful for this comment, as it allowed us to discover a collation error in our post-processing methods which affected the tabulation of our results, acutely obvious for East Asia and the Pacific and well as South Asia. This directly affects Table 1, which has been revised with correct values. After scrutinous checking, it appears this error only impacts this specific table and, consequently, the adjoining text describing it. The corrected Table 1 (including RCP/SSP ranges mentioned in Response 27) is shown below with updated caption:

"Table 1: Current (2020) and future (2080) EAD (Billion USD) and EAAP (Millions) per World Bank analytical region. The stand-alone future EAD and EAAP values are provided assuming RCP 6.0 / SSP2, which the ranges for future EAD and EAAP are provided across all RCP / SSP combinations, specifically the aforementioned combination as well as RCP 2.6 / SSP1 and RCP 8.5 / SSP3."

| World Bank Analytical Region | 2020 EAD | 2080 EAD | 2080 EAD Range | 2020 EAAP | 2080 EAAP | 2080 EAAP Range |
|---|---|---|---|---|---|---|
| East Asia & Pacific | 1.885 | 515.2 | 294.8 – 693.8 | 1.942 | 4.735 | 3.704 – 7.193 |
| Europe & Central Asia | 0.510 | 22.61 | 14.68 – 25.21 | 0.087 | 0.138 | 0.116 – 0.138 |
| Latin America & Caribbean | 0.309 | 17.83 | 14.94 – 19.80 | 0.057 | 0.119 | 0.084 – 0.237 |
| Middle East & North Africa | 0.130 | 33.07 | 18.62 – 40.21 | 0.080 | 0.243 | 0.206 – 0.356 |
| North America | 0.117 | 34.31 | 24.93 – 34.31 | 0.039 | 0.108 | 0.064 – 0.109 |
| South Asia | 0.179 | 187.7 | 89.45 – 297.5 | 0.979 | 3.864 | 3.024 – 6.509 |
| Sub-Saharan Africa | 0.938 | 542.8 | 195.9 – 833.7 | 0.444 | 2.356 | 1.850 – 3.571 |
| GLOBAL TOTAL | 4.068 | 1354 | 653.4 - 1941 | 3.628 | 11.56 | 9.112 – 18.05 |

Relevant values mentioned throughout the text have been checked and revised accordingly, including in the abstract and conclusion of the manuscript.

With this correction, it should be noted that some discrepancies should still be expected between our work and that of Tiggeloven et al. (2020), even though that study serves as a basis for the methodology of our manuscript. We both present similar, but distinct, "Middle of the Road" scenarios (RCP 4.5 / SSP 2 in Tiggeloven et al. (2020) and RCP 6.0 / SSP 2 in ours). In addition to the expected differences between these two different RCP/SSP combinations, here we use of updated hazard and more detailed exposure datasets to provide a more accurate and precise depiction of coastal flood risk.

**Comment 31: Line 374 – Why? What did you consider to result in these numbers?**

Response 31: This stems from the notion mentioned in comment 19: these three DRR measures are spatially limited to areas where ecosystem or urban constraints are met, depending on the DRR measure considered. Meanwhile, dyke and coastal levee modification can be performed in all regions with future coastal flood risk based on the assumptions we have set in our methodology. For example, in 213 sub-national

regions salt marshes or (restorable) mangroves were present, meaning in the remaining 714 sub-national regions this DRR measure is not applicable. Text has been added at this line for further clarification.

**Comment 32: Review your references: Wrong years in the citations. For example, line 53 references Hinkel and Klein (2014). There is only Hinkel and Klein (2009) in the reference list.**

Response 32: This in-text reference has been corrected to Hinkel and Klein (2009), with remaining in-text references checked for correctness.

**Comment 33: Figure 1 could have a better, more descriptive caption.**

Response 33: The caption for Figure 1 is expanded to the following:

> "Figure 1: Our study's analysis follows three main methodological steps consisting of estimating current and future coastal flood risk (labelled as "2.1 Risk Estimation"), modelling of four distinct DRR measures ("2.2 Risk Reduction"), and evaluating said measures in terms of their efficacy and economic performance ("2.3 Measure Evaluation"). Subsequent numbering in the figure refers to the relevant subsection of text. The literature referenced in the figure denotes the source of data or methodology used (as described in further detail in subsequent sections), while the colours represent different components of risk addressed."

**Comment 34: Line 385 – the largest cost is for Dry-proofing, not dykes. Please correct.**

Response 34: Thank you for this catch! This line has been corrected as follows:

> "The largest overall benefit of the evaluated DRR measure is for dykes and coastal levees, while the most costly is dry-proofing."

**Comment 35: Line 400 – Here you say "Over 90% of all sub-national regions can meet their respective relative-risk constant targets using at least one of the measures", but in line 417, you say, "the results show that implementing dykes and coastal levees has a global efficacy of 98% towards achieving relative-risk constant targets". How did you get to the "90%" number? It's confusing that the global analysis results in 98% and the sub-national one in 90%. Please clarify.**

Response 35: We apologize for this confusion, as we have inadvertently interchanged allusions to reduction amounts aggregated on the global scale and the number of sub-national areas where regional risk targets are achieved. To refer to the specific point raised above, 98% of the global risk reductions needed are achieved by dykes and levees. Meanwhile, in 90% of sub-national regions (835), the regional risk target can

be fully achieved. By extension, this means the remaining 2% of global risk reductions needed that are not achieved are spread amongst 92 (or 10% of) sub-national regions.

The text has been thoroughly examined to distinguish between these two interrelated, but distinct, descriptors of our results.

**Comment 36: Line 401 – 160 regions. What is the percentage considering the 90%?**

Response 36: This amounts to roughly one-fifth of all sub-national regions which we have estimated will experience future coastal flood risk. This has been clarified in the text of the manuscript's final paragraph.

**Comment 37: Line 458 – $544 million is coming from where?**

Response 37: We thank the referee for catching this unintended textual error. The actual value of costs is $366 million, as listed in Table 2.

References added to the revised manuscript

Meyer, V., Haase, D., & Scheuer, S. (2009). Flood risk assessment in European river basins—concept, methods, and challenges exemplified at the Mulde river. *Integrated environmental assessment and management*, 5(1), 17-26.

Torio, D. D., & Chmura, G. L. (2013). Assessing coastal squeeze of tidal wetlands. *Journal of Coastal Research*, 29(5), 1049-1061.